

# Growth of ice particle mass and projected area during riming

**Ehsan Erfani[1,2] and David L. Mitchell[1]**

[1] {Desert Research Institute, Reno, Nevada, USA}

[2] {Graduate Program in Atmospheric Sciences, University of Nevada, Reno, Nevada, USA}

Correspondence to: Ehsan Erfani (Ehsan@nevada.unr.edu)

Key points:

Rimed particle projected area- and mass-dimension expressions are developed and validated.

A convenient means of relating the unrimed and rimed $m$-$D$ and $A$-$D$ expressions was developed.

Equations are provided to calculate collision efficiency for use in models.





**Abstract**
There is a long-standing challenge in cloud and climate models to simulate the process of ice
particle riming realistically, partly due to the unrealistic parameterization of the growth of ice
particle mass ($m$) and projected area ($A$) during riming. This study addresses this problem, utilizing
ground-based measurements of m and ice particle maximum dimension ($D$) and also theory to
formulate simple expressions describing the dependence of $m$ and $A$ on riming. It was observed
that $\beta$ in the $m$-$D$ power law $m = \alpha D^{\beta}$ appears independent of riming before the formation of
graupel, with $\alpha$ accounting for the ice particle mass increase due to riming. This semi-empirical
approach accounts for the degree of riming and renders a gradual and smooth ice particle growth
process from unrimed ice particles to graupel, and thus avoids discontinuities in $m$ and $A$ during
accretional growth. The treatment for riming is explicit, and includes the parameterization of the
ice crystal-cloud droplet collision efficiency ($E_c$) for hexagonal columns and plates using
hydrodynamic theory. In particular, $E_c$ for cloud droplet diameters less than 10 μm are estimated,
and under some conditions observed in mixed phase clouds, these droplets can account for roughly
half of the mass growth rate from riming. These physically-meaningful yet simple methods can be
used in models to improve the riming process.
Keywords: ice cloud microphysics, ice particle growth, riming, collision efficiency, cloud models,
climate models





## 1    Introduction

Observational studies have determined that the riming process contributes substantially to snowfall rates. Along the coastal plains of northern Japan, riming was responsible for 50% to ~100% of the mass in snow collected at ground level, which included graupel particles (Harimaya and Sato, 1989). When only snowflakes were considered (no graupel), riming contributed between 40% and 63% of the snow mass. In the Colorado Rocky Mountains, Feng and Grant (1982) found that, for the same number flux, the snowfall rate for rimed plates and dendrites was about twice the snowfall rate for unrimed plates and dendrites (implying that about half of the snowfall rate was due to riming). In the Sierra Nevada mountains of California, Mitchell et al. (1990; hereafter M90) estimated that riming contributed 30% to 40 % of the mass of fresh snow during two snowfall events. Thus, an improved treatment of the riming process in cloud resolving models could significantly improve predicted snowfall amounts. This could also translate to improved quantitative precipitation estimates (QPE) from National Weather Service radar systems during winter. For example, a simple snow growth model (SGM) can be coupled with NWS radar reflectivity as described in Mitchell et al. (2006) to improve QPE, and adding the riming process should further improve these QPEs during winter storms.

The life cycle of Arctic mixed phase clouds, which strongly affect the Arctic energy budget and climate, should be affected by the ice mass flux ($M_f$) at cloud base (representing a moisture sink). Riming has a strong impact on ice particle fallspeeds (Mitchell, 1996; hereafter M96), and $M_f$ can be estimated as $M_f$ = IWC $V_m$, where $V_m$ is the mass-weighted fallspeed at cloud base and IWC is the ice water content. Since riming strongly contributes to both IWC and $V_m$, it has a powerful impact on $M_f$.

### 1.1    Characteristics of Riming

Riming (accretion of supercooled water droplets on ice particles) occurs in mixed-phase clouds where ice particles and water droplets coexist at temperatures ($T$) between -37.5 °C and 0 °C in convective clouds in the Tropics (Rosenfeld and Woodley, 2000; Mitchell and d'Entremont, 2012), and at -40.5 °C < $T$ < 0 °C in wave clouds over continental mountains (Heymsfield and Miloshevich, 1993). Mixed-phase clouds are persistent in both the Arctic and in tropical regions, as they happen nearly half of the time in the western Arctic (Shupe et al., 2006) and they





contribute to tropical convective storms having large amounts of supercooled water (Rosenfeld and
Woodley, 2000). They also constitute a large portion of the cloud fraction in mid-latitude storm
tracks (e.g. Hobbs, 1978; Matejka et al., 1980). However, a lack of observations in mixed-phase
clouds (resulting from the challenge of detecting layers of supercooled liquid water in the ice-
dominated parts of clouds) impeded an accurate computation of the liquid water content (LWC) to
IWC ratio, which therefore limits an understanding of riming (Kalesse et al., 2016). Wind tunnel
experiments by Takahashi and Fukuta (1988) and Fukuta and Takahashi (1999) measured the
riming enhancement as an increase in ice particle fallspeed ($V$). They also showed that riming has a
peak at -10.5 °C, where ice particles are isometric, and therefore have higher $V$.
The wind tunnel experiment of Pflaum *et al.* (1979) showed that a cone-like graupel forms, when
riming occurs on the bottom side of a falling planar crystal. However, if the particle flips over
during fallout, a lump graupel forms ultimately. Heymsfield (1982) developed a parcel model, and
demonstrated that growth of ice crystals by riming process occurs on their minor axis, and
therefore they evolve to graupel with spherical shape of the same dimension. In this model,
accreted mass fills in the unoccupied volume of the ultimately spherical graupel via riming growth.
In this way, ice particle mass increases while ice particle maximum dimension is conserved. The
increase in dimension due to riming initiates once the ice particle obtains a spherical shape. This
method was employed by several models to represent riming (Morrison and Grabowski, 2008;
hereafter MG08; Morrison and Grabowski, 2010; Jensen and Harrington, 2015; hereafter JH15;
Morrison and Milbrandt, 2015).
Many studies have developed ice particle mass-dimension (*m-D*) power law relationships for
specific ice particle shapes or environmental conditions, which have the form:

$$m = \alpha D^{\beta} , \tag{1}$$

where $\alpha$ is prefactor, and $\beta$ is power exponent, and both are constants over a specific size range.
They are determined via direct measurements of ice particle mass and dimension (Locatelli and
Hobbs, 1974; M90), or are constrained through aircraft measurements of the ice particle size
distribution (PSD) and IWC (Heymsfield et al., 2010; Cotton et al., 2012). Similar power laws
have been developed for projected area-dimension (*A-D*) relationships:



$$A = \gamma D^{\delta}, \tag{2}$$

where $\gamma$ and $\delta$ are constants over a specific size range derived by direct measurements of ice
particle projected area and dimension (M96). When comparing rimed particles with the same size,
lump graupel has the largest mass and area relative to cone-like graupel or hexagonal graupel, and
densely rimed dendrites have still lower values (Locatelli and Hobbs, 1974; M96). The $m$-$D$ and $A$-
$D$ power laws are dependent on the size range considered, and it often takes two or even three $m$-$D$
power laws to describe a given $m$-$D$ relationship over all relevant sizes. To address this issue,
Erfani and Mitchell (2016; hereafter EM16) developed a single $m$-$D$ and $A$-$D$ second-order
polynomial curve fit in log-log space for 20 μm $\leq D \leq$ 4000 μm for each cloud type (synoptic or
anvil) and temperature range. Such expressions can easily be reduced to power laws for use in
models and remote sensing, and provide size-dependent power law coefficients ($\alpha$, $\beta$, $\gamma$ and $\delta$). For
this reason, they are useful for characterizing a gradual change in power law coefficients with ice
particle growth.
Since explicit modeling of the riming process is computationally expensive, graupel and hail
categories were not considered in some bulk microphysics parameterizations used in some global
climate models or GCMs (Morrison and Gettelman, 2008; Gettelman and Morrison, 2015). The
common ice microphysics approach in most cloud and climate models is the separation of ice into
various hydrometeor categories such as cloud ice, snowflakes, and graupel (Rutledge and Hobbs,
1984; Ferrier, 1994; Fowler et al., 1996; Reisin et al. 1996; Morrison and Gettelman, 2008;
Gettelman and Morrison, 2015). The transition between various hydrometeors occurs by
autoconversion from one hydrometeor to another. However, such autoconversion is arbitrary and
poorly constrained, and as shown by Eidhammer et al. (2014), cloud radiative properties were
sensitive to the choice of autoconversion threshold size in the Community Atmosphere Model
version 5 (CAM5). This is because the distinct boundaries between various ice hydrometeor
categories impose abrupt microphysical changes, while in nature the transition processes are
gradual. To overcome this problem, MG08 advanced a bulk model that employed vapor diffusion
and the riming processes, and used multiple $m$-$D$ and $A$-$D$ power laws (Eqs. 1 and 2) to
characterize ice particles associated with different parts of the PSD. This method was applied to a
bin model developed by Morrison and Grabowski (2010), and was later used in a four-moment
bulk model that also included the process of ice particle aggregation (Morrison and Milbrandt,





2015). Such *m-D* and *A-D* expressions resulted in a smooth transition from crystal mass to graupel
mass (continuous *m-D* expressions over the PSD). However, discontinuities were observed in
transition between various *A-D* expressions over the PSD. JH15 developed a detailed ice growth
model that simulates ice particle habit and mass via the processes of vapor deposition and riming.
This model is also a single-category scheme, but it does not employ *m-D* and *A-D* power laws;
instead, it computes the growth of ice particles along the major and minor axes of oblate or prolate
spheroids (representing hexagonal plates or columns). Therefore, the model is able to simulate
simple ice particle shapes, and also captures the temperature-dependency of vapor deposition and
the riming processes (since particle shape is a function of temperature). The simulated ice particle
shape, mass, and fallspeed are in good agreement with observational data from wind tunnel
experiments on ice crystal growth.

### 1.2   Collision Efficiency

One important factor in the modeling of riming is the calculation of the collision efficiency ($E_c$)
between ice particles and cloud droplets (Pruppacher and Klett, 1997). $E_c$ was calculated as a
function of ice particle $D$ and cloud droplet diameter ($d$) via both experimental measurements
(Sasyo and Tokuue, 1973, hereafter ST73; Kajikawa, 1974, hereafter K74; Murakami et al., 1985)
and theoretical/numerical calculations (Beard and Grover, 1974; Pitter and Pruppacher, 1974;
Schlamp, 1975; Pitter, 1977; Wang and Ji, 2000, hereafter WJ00). The difference in $E_c$ between
various studies is due to the strong sensitivity of $E_c$ to the ice particle shape as well as the
assumptions and limitations in different studies. Experimental measurements of $E_c$ have been
conducted in vertical wind tunnels. Such studies are rare due to the difficulty and limitations of
experiments, and were limited to only planar ice crystals or circular disks with $D > 1$ mm
(Reynolds number or Re > 40). Murakami et al. (1985) studied the $E_c$ between polystyrene latex
spheres ($d < 6$ μm) and planar ice crystals (1.5 mm < $D$ < 5 mm, and 70 < Re <300) at their free
fallspeeds. ST73 investigated fixed hexagonal plates (5 mm < $D$ < 20 mm) that are exposed to
water droplets contained in airflow in a vertical wind tunnel. Although $d$ ranges from 19 μm to 41
μm, more than 80% of droplets had $d$ between 20 μm and 25 μm. K74 measured $E_c$ via collection
of water droplets (2.5 μm < $d$ < 17.5 μm) by freely-falling particles (both natural snow crystals and
ice crystal models made of non-water substance) of various shapes (e.g. circular disks, hexagonal
plates and broad-branched plates) with Re < 100 in a wind tunnel. Numerical studies calculate the



flow field around particles by solving the Navier-Stokes equation via numerical methods. The
challenges for numerical studies are the complex shapes of ice crystals as well as the effect of
turbulence. Early studies assumed steady state flow with simplified shapes such as an oblate
spheroid with $2 \leq \text{Re} \leq 20$ as an approximation for planar crystals (Pitter and Pruppacher, 1974;
Pitter, 1977), and an infinite cylinder with $0.2 \leq \text{Re} \leq 20$ as an approximation for columnar crystals
(Schlamp, 1975). The main difference in $E_c$ between experimental and numerical studies is
observed for small droplets ($d < 10$ µm), where numerical $E_c$ is zero in this range, but the
experimental results indicate finite $E_c$. As explained by K74, this difference might be due to the
assumption of a steady flow field around the ice particle in the early numerical studies. WJ00
developed a numerical model of 3-D non-steady flow around pristine crystals (such as hexagonal
plates with $1 \leq \text{Re} \leq 120$ and columnar crystals of finite length and with $0.2 \leq \text{Re} \leq 20$) and water
droplets ($d < 200$ µm). Contrary to early numerical studies and in agreement with experimental
results, they showed that $E_c$ for small droplets has finite values for hexagonal plates (hexagonal
columns) with $\text{Re} \geq 10$ ($\text{Re} \geq 0.2$).
Due to its expensive computation, $E_c$ is sometimes assumed to be constant in the models (e.g., $E_c =$
0.75 in MG08; $E_c = 1$ in Rutledge and Hobbs, 1984; Ferrier, 1994; Fowler et al., 1996; Morrison
and Milbrandt, 2015). Hall (1980; hereafter H80) provided an equation for $E_c$ representative of
hexagonal plates by fitting ellipse curves to the data of Pitter and Pruppacher (1974) and Pitter
(1977). Although this relationship is practical and was used by several models (Morrison and
Grabowski, 2010; JH15; Kalesse et al, 2016), it has limitations due to the natural shortcomings of
the original numerical studies (assumptions of steady flow, ice oblate spheroids with $\text{Re} < 20$ as an
approximation for hexagonal plates, water droplets with $d < 20$ µm, and zero $E_c$ for $d < 10$ µm).
WJ00 improved the computation of $E_c$ by solving these issues, but did not provide an equation for
use in the models. JH15 modified the equation from Beard and Grover (1974) for spherical
raindrops in steady flow, and calculated $E_c$ between prolate spheroids (as an approximation for
hexagonal columns) and small water droplets. $E_c$ calculated in this way compares well with the
numerical study of WJ00 for $5$ µm $< d < 20$ µm.
Another challenge exists in the calculation of $E_c$ between graupel and cloud droplets. Most studies
used $E_c$ from Beard and Grover (1974), and therefore assumed that this $E_c$ is equal to the collision
efficiency between raindrops and water drops (Reisin et al. 1996; Pinski et al. 1998; Khain et al.



1999; Morrison and Grabowski, 2010). The justification for this assumption was the similar shape
between graupel and raindrops. However, such particles have different natural features (e.g.,
density and surface roughness). To solve this issue, Rasmussen and Heymsfield (1985) suggested
that $E_c$ between graupel and cloud droplets can be calculated by modification of the results of
Beard and Grover (1974) for $E_c$ between raindrops and water droplets. On the other hand, von
Blohn et al. (2009) investigated experimental $E_c$ between freely falling spherical ice particles
(initially 580 µm $< D <$ 760 µm) and water droplets (20 µm $< d <$ 40 µm) in a vertical wind tunnel
with laminar flow. They showed that collection kernels of ice particles are higher than that of
raindrops, and therefore calculated a correction factor to account for the error in $E_c$, when
assuming raindrops instead of graupel.
The objective of this study is to develop various empirical and theoretical approaches to represent
the continuous and gradual growth of ice particle mass and projected area during riming in a
realistic and yet simple way, suitable for models. Section 2 of this study explains the data and
method. In Sect. 3, results from a ground-based field campaign are applied to investigate $m$-$D$
relationships during riming. Section 4 introduces a method to parameterize riming. In Sect. 5, new
practical equations are presented to calculate $E_c$ for hexagonal plates and hexagonal columns.
Calculations of the mass growth rate due to riming are given in Sect. 6, and conclusions are
provided in Sect. 7.

## 2    Data and methods

Ground-based direct measurements of $m$ and $D$ from Sierra Cooperative Pilot Project (SCPP; see
M90) during winter storms in Sierra Nevada Mountains are utilized in this study. SCPP was a field
campaign on cloud seeding from 1986 to 1988, and for one part of that project, natural ice particles
were collected during snow storms in a polystyrene petri dish and then the particles were
photographed using a microscope equipped with a camera. Then a heat-lamp was used to melt
these ice particles, and immediately after melting another photograph was taken of the hemispheric
water drops (contact angle on polystyrene = 87.4 degrees). The images were used later in the lab to
measure the maximum dimension ($D$) of individual ice particles (defined as diameter of a
circumscribed circle around the particle). Also, the diameter of the water hemispheres was



measured, and from this the volume and mass of individual ice particles were computed. Also indicated were individual ice particle shapes (if recognizable), basic level of riming (e.g., light, moderate, heavy riming, or graupel), and temperature range in which the observed ice particle shape originated. Software was developed to extract all combinations of particle shapes (for a detailed explanation of sampling and measurements, see M90).

EM16 provided *m-D* curve fits based on Cloud Particle Imager (CPI) measurements from the Department of Energy (DOE)-Atmospheric Radiation Measurement (ARM) funded Small Particles In Cirrus (SPartICus) field campaign for $D < 100$ μm and a subset of SCPP data for $D > 100$ μm. This subset of SCPP includes only unrimed ice particles that have habits identical to those in cirrus clouds (selected based only on ice particles that have habits formed in the temperature range between -40°C and -20°C). There are 827 ice particles that are categorized in this subset. Hereafter, this subset of SCPP is referred to as "cold habit SCPP". The SCPP data has a total of 4869 ice particles, consisting of 2341 unrimed or lightly-rimed particles (such as plates, dendrites, columns, needles, bullets, bullet rosettes, side planes, and aggregates and fragments of these shapes), 1440 moderately- or heavily-rimed particles (such as rimed plates, rimed dendrites, rimed columns, and graupel), and 1088 unclassified particles. There were 118 unrimed dendrites, including ordinary, stellar and fern-like dendrites, classified using the Magono and Lee scheme (Pruppacher and Klett, 1997) as P1e, P1d and P1f, as well as fragments and aggregates of these shapes. 80% of unrimed dendrites were P1e. Columnar crystals consisted of 262 N1e (long solid columns) and 337 C2b (combination of long solid columns) crystals. Some ice crystals classified as unrimed may be lightly rimed due to limitations in the magnification used. Moreover, 852 particles were classified as heavily rimed dendrites, consisting of graupel-like snow of hexagonal type (R3a), graupel-like snow of lump type (R3b), and graupel-like snow with nonrimed extensions (R3c), of which 99% were R3b. These correspond to heavily rimed dendrites having graupel-like centers but with rimed branches extending outwards revealing the dendritic origin. Also classified were total of 67 lump graupel (R4b), cone-like graupel (R4c), and hexagonal graupel (R4a); R4b and R4c are graupel with non- discernable original habit, whereas R4a forms just prior to R4b or R4c, with its hexagonal origin still recognizable.

In order to represent the natural variability of ice particle mass, all identifiable particles are initially shown with their actual mass and maximum dimension. Thereafter, to quantify the





variability and to further investigate *m-D* power laws and the rimed-to-unrimed mass ratio, the ice
PSDs were divided into size bins with intervals of 100 μm between 100 and 1000 μm, and with
subsequent intervals of 200, 200, 400, 600, 600 and 1000 μm (up to 4000μm) at larger sizes to
supply sufficient sampling numbers in each size bin. In order to investigate the riming effect, all
identifiable particles are divided into rimed and unrimed categories: unrimed or lightly-rimed ice
particles were classified in unrimed category, whereas moderately- or heavily-rimed particles were
considered in rimed category.

## 3    Measurements of ice particle mass and dimension in frontal clouds

The purpose of this section is to investigate how the CPI and cold habit SCPP curve fit from EM16
compares with all the SCPP data, since this could indicate how representative this curve fit is for
ice particles found in Sierra Nevada frontal clouds. This comparison is shown in Fig. 1a for all ice
particles that could be classified (3781 ice particles). The curve fit appears to bisect the data well.
It is also seen that rimed ice particles tend to have larger mass on average, compared to unrimed
ice particle of the same size. Also displayed are the *m-D* power law expressions from Cotton et al.
(2012) and Heymsfield et al. (2010) that were acquired from synoptic ice clouds for -60 °C < *T* < -
20 °C and from both synoptic and anvil ice clouds for -60°C < *T* < 0°C, respectively. The grey
line, corresponding to spherical particles, serves as an upper limit to ice particle mass. The Cotton
et al. (2012) expression is composed of two power laws and accompanies the EM16 curve fit
significantly well for *D* > 100 μm, with differences in mass that never exceed 50%. The
Heymsfield et al. (2010) expression is based on a single power law and also estimates the curve fit
well, except for the size ranges *D* > 1000 μm and *D* < 100 μm ,where the differences in mass can
extend to about 100%. Figure 1b displays the EM16 curve fit along with all SCPP data (including
those that could not be classified), where the ice PSDs were divided into size bins, as explained in
Sect. 2. In this way, mean *D* and *m* in each size bin, and also the standard deviation (*σ*) in each size
interval for *D* and *m* are shown. Figure 1b shows that the curve fit is well within the *σ* of SCPP
mass and is mostly adjacent to the mean *m* for all size bins. The same is valid for the Cotton et al.
(2012) mass when the line is extrapolated to *D* > 500μm. The Heymsfield et al. (2010) line is only
within the *σ* of SCPP for 250 μm < *D* < 1400 μm. In order to be even more quantitative, the



percent difference between the SCPP mean ice particle mass in each size-bin of Fig. 1b and the corresponding mass from the cold habit SCPP curve fit from EM16 are computed (figure not shown). For $D > 200$ μm, percent differences are no more than 22%, with the curve fit slightly overestimating masses for $D > 1000$ μm. This agreement might result partially from the riming of the planar ice crystals and aggregates thereof (adding mass with little change in size) and partially from an abundance of unrimed and rimed high density compact ice particles. Indeed, 38% of the ice particles were moderate-to-heavily rimed. Based on the planar ice particles in this dataset (excluding side planes), we found that riming contributed to roughly 20-30% of ice particle mass on average for $D > 700$ μm, when riming was moderate-to-heavy. To summarize, it appears that the synoptic ice cloud curve fit for -40 °C $< T \leq$ -20 °C provides a realistic bulk estimate for ice particle masses in frontal clouds.

## 4 Parameterization of riming

### 4.1 Dependence of *β* and *α* on riming

A long-standing problem in cloud modeling is the treatment of $\alpha$, $\beta$, $\gamma$ and $\delta$ as a function of ice particle riming. Since riming leads to graupel formation and graupel tends to be quasi-spherical, it is intuitive to assume that β and δ will approach limiting values of 3 and 2, respectively (corresponding to ice spheres), as more and more supercooled liquid water is accreted by an ice particle to produce graupel. One common approach in many cloud models (that use an *m-D* relationship) is to assume that β is equal to ~ 2 for unrimed crystals and is equal to ~ 3 for graupel. This implies that riming enhances β. This assumption is tested in this section by using SCPP data with the objective of developing observational-based guidelines for modeling the process of riming. To test this assumption for β, the size-resolved masses of rimed and unrimed ice particles from the same basic shape category are needed. In this section, we used heavily rimed dendrites (R3a, R3b and R3c) and unrimed dendrites (P1e, P1d and P1f). In addition, this data was partitioned into the same size-intervals described earlier to calculate the mean *m* and *D* in each size-interval for unrimed and heavily rimed dendrite crystals, along with their $\sigma$. All these results are shown in Fig. 2. Size-intervals having less than 3 measurements are not represented. Most of the data for unrimed crystals is associated with $D > 600$μm. One can see quantitatively how the





mean masses for rimed dendrites are substantially greater than those for unrimed dendrites on
average for the same size-interval, in agreement with the hypothesis of Heymsfield (1982).
Using only the size-intervals containing at least 3 measurements, the *m-D* power law for the
unrimed dendrites is:

$$m = 0.001263 D^{1.912} \, , \tag{3}$$

and for heavily rimed dendrites is:

$$m = 0.001988 D^{1.784} \, , \tag{4}$$

where all variables have cgs units. If the size-interval corresponding to the largest unrimed
dendrites is not used in the least-square fit calculation, the *m-D* expression for unrimed dendrites
becomes:

$$m = 0.0009393 D^{1.786} \, , \tag{5}$$

having an exponent nearly identical to that in Eq. (4). It is now apparent that the traditional
hypothesis that $\beta$ increases with riming is not correct, at least not for these measurements. This can
be understood by noting that $\beta$ does not necessarily indicate the morphology of an ice particle
within a given size-interval, but rather indicates the mass rate-of-change with respect to size (since
$\beta$ is the slope of the *m-D* line in log-log space). This can also be seen qualitatively in Fig. 2, where
the rimed and unrimed data points represent the same slope for the *m-D* line in log-log space. In
addition, the *m-D* power law for lump graupel and cone-like graupel has the form of
$m = 0.0078 D^{2.162}$ that represents a slight increase in $\beta$ for graupel which is significantly less than
spherical $\beta$ (which is equal to 3). All these observations are in agreement with the experiment of
Rogers (1974) in which $\beta$ was similar for unrimed and rimed snowflakes. The results of Rogers
(1974) were used in the modeling work of MG08 and Morrison and Grabowski (2010) to assume
that riming does not change $\beta$ for planar ice crystals. Morrison and Milbrandt (2015) used a similar
assumption based on the observations of Rogers (1974) and Mitchell and Erfani (2014), and they
explained that the reason for the conservation of $\beta$ during riming is the fact that ice particle
maximum dimension $D$ does not significantly change by riming while $m$ does increase





significantly. A similar assumption is also valid for hexagonal columns. The impact of moderate to
heavy riming on $\beta$ for hexagonal columns was demonstrated in M90 (see their Table 1 and Sect.
4d). For these columnar crystals, riming had no effect on $\beta$ (i.e., $\beta$ was 1.8 for both rimed and
unrimed columns), indicating that riming can be modeled by only increasing $\alpha$ for these crystals.
Thus, it appears justified to treat $\beta$ as constant during the riming process for both dendritic and
columnar ice crystals:

$$\beta = \beta_u,\tag{6}$$

where subscript $u$ denotes unrimed conditions. The IWC is defined as:

$$\text{IWC} = \int m(D)n(D)dD = \alpha \int D^{\beta} n(D)dD\tag{7}$$

where $n(D)$ is number density. We explained that $\beta$ and $D$ do not change during riming. Also
unchanged is $n(D)$, because it is a function only of $D$, and the number of ice particles in each size
bin is not affected by riming. Therefore, the dependence of $\alpha$ on riming can be calculated by
knowing the contribution of riming to the IWC:

$$\frac{\alpha}{\alpha_u} \approx \frac{\text{IWC}}{\text{IWC}_u}.\tag{8}$$

Note that riming occurs only when ice particles have a $D$ greater than the riming threshold size
($D_{\text{thres}}$: the smallest ice crystal $D$ for which riming can occur). Early observations (Harimaya, 1975)
and numerical studies (Pitter and Pruppacher, 1974; Pitter, 1977) determined a $D_{\text{thres}}$ being around
300 µm. However, it was later shown by both observational (Bruntjes et al., 1987) and numerical
(WJ00) studies that such $D_{\text{thres}}$ is around 35 µm, 110 µm, and 200 µm for hexagonal columns,
hexagonal plates, and broad-branched crystals, respectively (note that all these dimensions are
along a-axis of crystals).
Since $\beta$ is essentially the same in Eqs. (4) and (5), their prefactor ratio ($\alpha$ in Eq. 4 divided by $\alpha$ in
Eq. 5, which is equal to 2.12) indicates that riming contributed slightly more than half the mass of
the rimed dendrites. This can be confirmed by calculation of the ratio of mean rimed dendrite mass
($m_r$) to mean unrimed dendrite mass ($m_u$) for each common size-interval, as shown in Fig. 3. This
riming ratio ($m_r/m_u$) for each size-bin varies from ~ 0.5 to 3 with many values close to 2. The





weighted average of $m_r/m_u$ is equal to 2.0, supporting the first estimate of 2.12. The largest
deviation from the mean for 300 µm $< D <$ 400 µm may be due to only a single unrimed ice crystal
of anomalous mass in this size bin.
Equations (4) and (5) also suggest a means of adapting the $m$-$D$ curve fit in Fig. 1 for modeling the
riming process in mixed phase clouds. Since this curve fit is representative of ice particle
populations in frontal clouds (containing a mixture of unrimed and rimed particles), it can be
adapted for modeling the riming process in frontal clouds. Since $\beta$ should be essentially the same
for both unrimed and the mixture of unrimed plus rimed SCPP ice particles, the ratio of their
corresponding prefactors (i.e. $\alpha_u/\alpha_{mix}$) can be multiplied by the mass predicted by the curve fit
equation to yield masses appropriate for unrimed particles. For the ice particles plotted in Fig. 1a,
$m_u/m_{mix}$ is equal to 0.650 (where $m_{mix}$ includes all these particles and $m_u/m_{mix}$ was calculated by the
same method that calculated $m_r/m_u$ in Fig. 3). This implies that multiplying the mass predicted by
the curve fit in Fig. 1 by a factor of 0.65 will yield masses proper for unrimed ice particles. To
model the riming process in frontal clouds, these unrimed particles can be subjected to the riming
growth equations described below as well as Eq. (8).
## 4.2   Dependence of $\delta$ and $\gamma$ on riming
Since there are no SCPP $A$-$D$ measurements that correspond with the $m$-$D$ measurements used in
Sect. 4.1, a purely empirical evaluation of the dependence of $\delta$ and $\gamma$ on riming was not possible.
However, Fontaine et al. (2014) simulated numerous ice particles (pristine crystals, aggregates,
and rimed particles) with various 3-D shapes and also their projected area (assuming random
orientation). By this, they were able to develop a linear expression between $\beta$ and $\delta$. This linear
expression implies that $\delta$ is constant during the riming process, since $\beta$ has no riming dependency
(see Sect. 4.1):

$$\delta = \delta_u \tag{9}$$

The reason for this can be explained by noting that the riming process often affects $A$ but does not
change $D$ (by filling the space between ice particle branches) significantly prior to graupel
formation. This is also evident from observations, as shown in Table 1 of M96, where $\delta$ is equal to
2 for both hexagonal plates and lump graupel. For constant $\delta$, only $\gamma$ depends on riming, and to





express $\gamma$ as a function of riming, we developed a method that estimates the change in $A$ by riming
as a function of the change in $m$:

$$A = (A_{max} - A_u)R + A_u \qquad (10)$$

where $A_{max}$ is the maximum projected area due to riming (which is the graupel $A$), and $R$ is the
riming factor defined as:

$$R = \frac{m - m_u}{m_{max} - m_u} \qquad (11)$$

where $m_{max}$ is the graupel $m$ (having the same $D$ as $m$ and $m_u$). $R$ is between 0 and 1, with 0
denoting no riming and 1 indicating graupel formation. In other words, when an ice crystal is
unrimed, $m = m_u$ and $A = A_u$; and when $m = m_{max}$ and $A = A_{max}$, the ice crystal attains graupel
status. For a given $D$, $\gamma = A/D^{\delta}$, and in this way the riming dependence of $\alpha$ and $\gamma$ can be treated,
while $\beta$ and $\delta$ are independent of riming. Note that Eq. (10) assumes a linear relationship between
$m$ and $A$ during riming, an assumption that can be investigated through future research.

### 11   4.2.1   Planar ice crystals

Using the approach above, $m$ (in particular, $\alpha$) should first be determined as a function of riming
using conventional theory (this will be discussed in Sect. 6), and then Eqs. (8), (10) and (11) can
be applied to calculate $A$. In order to determine $m_{max}$, we calculated the $m_r/m_u$ that corresponds to
graupel (R4a, R4b, and R4c) and unrimed dendrites (P1d, P1e, and P1f), as shown in Fig. 4a.
Small variability is seen for $D < 1200$ μm (ranges from 3 to 3.8, with the exception of smallest size
bin), whereas large variability exists (between 1.6 and 8.4) for larger sizes due to the small number
of graupel in each size bin. The weighted average for this $m_r/m_u$ ratio is equal to 3.3 which can be
used to estimate $m_{max}$: $m_{max} \approx 3.3 \times m_u$ for dendrites. Since R4a occurs just before hexagonal
features are completely obscured by additional rime deposits, R4a graupel is ideal for estimating
$m_{max}$. Unfortunately there are only 14 R4a particles in the entire SCPP data set, with $D < 1200$ μm.
They exhibit a large variability in the $m_r/m_u$ ratio (ranging from 1.6 to 4.5) with a weighted average
of $m_r/m_u$ equal to 3.1 (figure not shown). Nonetheless the close agreement with the above $m_r/m_u$
ratio of 3.3 is encouraging. A similar observational analysis was conducted by Rogers (1974), who





found that $\alpha$ for heavily rimed snowflakes was 4 times larger than that for unrimed snowflakes
(and $\beta$ was similar for both rimed and unrimed snowflakes).
Since there is no observation to indicate $A_{\max}$, it can be approximated as the area of a circle having
the same $D$ ($A_{\mathrm{sphere}}$); but since graupel is not perfectly spherical, $A_{\max}$ can be better estimated as a
fraction of $A_{\mathrm{sphere}}$; $A_{\max} = kA_{sphere}$, where $k$ is correction factor. Heymsfield (1978) analyzed graupel
particles in northeastern Colorado, and found that their aspect ratio does not exceed 0.8. Using this
value, JH15 showed good agreement between their model and observational data from a wind
tunnel. Based on such analysis, $k$ is equal to 0.8. Further observational data are needed to
determine the value of $A_{\max}$ more accurately.
Once the graupel stage is attained, the graupel continues to grow through riming, and a different
methodology is required to describe riming growth at this growth stage, because graupel $D$
increases by riming. Once $m = m_{max}$, then a graupel bulk density is defined as:

$$\rho_g = \frac{m_{\max}}{V_g} \tag{12}$$

Where $V_g = (\pi/6)D_g^{\,3}$ and $D_g$ is graupel $D$ when $m = m_{max}$. For subsequent riming growth, $\rho_g$
remains constant. For this growth stage, riming does increase $D$ and $A$, which are determined as a
function of riming as:

$$D = \left( \frac{6m}{\pi\rho_g} \right)^{\frac{1}{3}} \tag{13}$$

$$A = k\frac{\pi}{4}D^2 \tag{14}$$

where $m$ is calculated as described in Sect. 6. As before, for a given $D$, $\gamma = A/D^\delta$, and in this way
riming growth is treated for all conditions.



**4.2.2   Columnar ice crystals**
Figure 4b represents $m_r/m_u$ between graupel (R4b and R4c) and unrimed columnar crystals (N1e
and N2c) in order to determine $m_{max}$ for columnar crystals. Relatively small variability of $m_r/m_u$
(between 1.6 and 3) is found for $D < 1400$ µm, with larger variability (from 1.4 to 9.4) found for
larger ice particles, with the weighted average of $m_r/m_u$ equal to 2.4, and therefore $m_{max} \approx 2.4 \times m_u$.
The higher variability for $D > 1400$ µm is likely due to a single graupel particle per size-bin.
**4.3   Testing the Baker and Lawson (2006) *m-A* expression with unrimed dendrites**
Some of the data shown in Fig. 2 describes an experiment investigating the ability of the Baker and
Lawson (2006) (hereafter BL06) *m-A* power law to reproduce the masses of unrimed dendrites that
presumably have relatively low area ratios (the ratio of the actual ice particle projected area to the
area of a circle having a diameter equal to the ice particle maximum dimension). A study by
Avramov et al. (2011) found that this power law overestimated the masses of low-density dendrites
(P1b), high-density dendrites (P1c), and low density dendrite aggregates, but that the BL06 power
law yielded masses consistent with high density dendrite aggregates at commonly observed sizes.
It is important to understand the potential limitations of this power law for dendrites due to their
abundance in Arctic mixed phase clouds and for the modeling of these clouds. Unfortunately, there
were only 7 unrimed and 2 lightly rimed dendrites in the BL06 dataset to investigate this finding.
These are represented in Fig. 2 by green circles; their masses were calculated from the BL06 *m-A*
expression using their measured projected areas. For $D < 1.4$ mm, the BL06 unrimed dendrite
masses are consistent with the unrimed dendrite masses from all SCPP data evaluated in this study
(e.g., are within $\pm 1\ \sigma$ of mean $m$ for each size-bin), but at larger sizes the BL06 unrimed dendrite
masses conform with rimed dendrite masses evaluated in this study. This suggests that for $D > 1.4$
mm, the BL06 *m-A* expression might overestimate the masses of unrimed dendrites by about a
factor of two. This is broadly consistent with Avramov et al. (2011) for the size range considered.
However, there is insufficient data here to draw any firm conclusions.
Although $A$ is more strongly correlated with ice particle $m$ than is $D$ (based on BL06), inferring $m$
or volume from a 2-D measurement is still ambiguous since different crystal habits exhibit
different degrees of ice thickness or volume for a given $A$. Thus, the BL06 *m-A* expression is not
expected to be universally valid for all ice crystal habits. On the other hand, when applied to $A$



measurements in cirrus clouds, it yields ice particle mass estimates that are very consistent with
two other studies that estimated *m-D* expressions for cirrus clouds (Heymsfield et al., 2010; Cotton
et al., 2012), as described in Sect. 3. In addition, a comparison with a cold-habit SCPP dataset
provided additional evidence that the BL06 *m-A* expression yields masses appropriate for ice
particles found in cirrus clouds. It also yields masses that are very consistent with the mean masses
obtained for all ice particles sampled during the SCPP, indicating that the BL06 *m-A* expression
appears representative of ice particle masses characteristic of Sierra Nevada snow storms. As
explained by EM16 and references therein, there is only about a 20% difference between IWCs
calculated from PSD using the BL06 *m-A* power law and collocated direct measurements of IWC
in tropical regions; however such differences can be as high as 100% in Polar Regions.

## 5 Collision Efficiencies

As mentioned in Sect. 1.2, there is a lack of practical methods in the literature for computing $E_c$ for
plates, columns, and graupel. In this section, equations are provided that calculate $E_c$ for hexagonal
plates and hexagonal columns, based on the data of WJ00. Such equations can be used in cloud
and climate models to treat the riming process.

### 5.1 Hexagonal plates

The numerical study of WJ00 is valid for unsteady flow, hexagonal ice plates with $1 < \mathrm{Re} < 120$
and $160\ \mu m < D < 1700\ \mu m$, and water droplets with $1\ \mu m < d < 100\ \mu m$. Re for hexagonal plates
is calculated based on the maximum dimension (e.g., $\mathrm{Re}_{plates} = DV/\varepsilon$, where $\varepsilon$ is kinematic
viscosity). Since there is not sufficient agreement between the historical H80 relationship and the
data of WJ00, we provided best fits to the data of WJ00 that has the form of:

$$E_c = \begin{cases} \left(0.787 K^{0.988}\right)\left(0.263 \ln \mathrm{Re} - 0.264\right), & 0.01 \le K \le 0.35 \quad \& \quad 2 < \mathrm{Re} \le 120 \\ \left(0.7475 \log K + 0.620\right)\left(0.263 \ln \mathrm{Re} - 0.264\right), & 0.35 < K \le K_{thres} \quad \& \quad 2 < \mathrm{Re} \le 120 \\ \sqrt{1 - \dfrac{1}{5}\left[\log(\dfrac{K}{K_{crit}}) - \sqrt{5}\right]^2}, & K_{thres} < K < 35 \quad \& \quad 1 \le \mathrm{Re} \le 120 \end{cases} \quad (15)$$

where $K$ is mixed Froude number of the system of water drop-ice particle, and is calculated as:

$$K = \frac{2(V-v)v}{Dg} ,$$  (16)

where $v$ is water drop fallspeed, and $g$ is gravitational acceleration. Since cloud water drops are in
Stokes regime, $v$ is calculated as the Stokes fallspeed (e.g., $v = g(\rho_w - \rho_a)d^2 / 18\mu$, where $\rho_w$ is
water density, $\rho_a$ is air density, and $\mu$ is dynamic viscosity), and $K$ is the same as the Stokes
number in this flow regime. $K_{\text{crit}}$ is the critical value of $K$ (where $E_c$ equals 0 in the third line in Eq.
15) and is expressed as a function of ice particle Re:

$$K_{crit} = \begin{cases} 1.250\,\text{Re}^{-0.0350}, & 1 < \text{Re} \leq 10 \\ 1.072\,\text{Re}^{-0.301}, & 10 < \text{Re} \leq 40 \\ 0.356\,\text{Re}^{-0.003}, & 40 < \text{Re} \leq 120 \end{cases} .$$  (17)

Based on Eq. (15), $E_c$ in the third line is physically meaningful only when $K \geq K_{\text{crit}}$. When $K < K_{\text{crit}}$,
$E_c$ in the third is imaginary and must be set to zero in order to avoid errors. $K_{\text{thres}}$ is the threshold of
$K$ between small and large cloud droplets, and is calculated based on Re in WJ00 as
$K_{thres} = -5.07 \times 10^{-10}\,\text{Re}^5 + 1.73 \times 10^{-7}\,\text{Re}^4 - 2.17 \times 10^{-5}\,\text{Re}^3 + 0.0013\,\text{Re}^2 - 0.037\,\text{Re} + 0.8355$, and has
values between 0.4 and 0.7. Alternatively, it can be calculated for a desired Re by equating $E_c$ from
the second line with $E_c$ from the third line in Eq. (15) (e.g., finding the intersection of curves
defined by the second and the third lines of Eq. 15) to avoid any discontinuity. The third line in Eq.
(15) is an ellipse fit similar to H80 equation, but such a fit cannot represent finite values of $E_c$ for
small drops (when $K < K_{\text{thres}}$), and therefore this ellipse fit is not valid for small drops. To
overcome this issue, curve fits are developed (the first and second lines in Eq. 15) similar to
Mitchell (1995; hereafter M95). M95 provided curve fits to experimental $E_c$ data described in
ST73, K74 and Murakami et al. (1985) that showed slight sensitivity to Re. Here, those equations
are modified and additional terms are employed to account for the Re dependence of $E_c$ for small
droplets, based on the data of WJ00.
The resulting curve fits for $E_c$ (Fig. 5a) show that the provided equations can represent the data of
WJ00 very well in various ranges of $K$ and Re. The percent error in $E_c$ between curve fits and




WJ00 data has a mean value of 6.65% with standard deviation of 3.67% for all Re and $K$. For a
given $K$, $E_c$ for planar crystals increases with an increase in Re because of the increase in the
plate's fallspeed. In addition, $E_c$ has a slight sensitivity to Re for Re $\geq$ 60. $E_c$ for small Re (Re $\leq$ 2)
appears to have a different pattern than that for larger Re, since $E_c$ has zero values for small water
drops ($K \leq 1$). This implies that smaller ice particles that have sizes slightly larger than the $D_{thres}$
are incapable of collecting the smaller drops. For a given Re, $E_c$ increases with increasing $K$,
associated with an increase in droplet diameter, but it does not exceed a value of unity. For
comparison, historical experiments by ST73 and K74 are also shown in this graph. K74 data for 10
$\leq$ Re $\leq$ 35 is in good agreement with the curve fit for Re = 10. Values of $E_c$ from K74 for 200 $\leq$ Re
$\leq$ 640 are slightly lower than curve fit for Re = 120. This does not seem to be a discrepancy,
because it is observed from the curve fits (based on WJ00) that $E_c$ is not sensitive to Re when Re $\geq$
60. This is also observed in K74 for large Re (their Fig. 14). $E_c$ from ST73 for Re = 97 is in good
agreement with the curve fit for $K \sim 1.5$, but is larger than the curve fit for $K \sim 0.3$. It is noteworthy
to explain the shortcomings of these experiments, as mentioned by Pruppacher and Klett (1997).
For the experiment of K74, when Re > 100, the flow is unsteady and leads to the eddy shedding
and formation of wakes at the top of the particle, which increases the uncertainty in fallspeed. For
the study of ST73, there is an extra problem: the air stream speed was not in agreement with the
fallspeed that the fixed collectors would have, if they were to fall freely.
For $K > 1.0$, M95 modified the relationship by Langmuir (1948) for $E_c$ between spherical water
raindrops and cloud droplets, and provided an expression as $E_c = (K+1.1)^2 / (K+1.6)^2$. However,
this relationship underestimates the best fits to the data of WJ00 (figure not shown). This confirms
the findings of von Blohn et al. (2009) who observed smaller $E_c$ for raindrops relative to graupel,
and highlights the need for using $E_c$ for ice particles with realistic shapes and avoiding $E_c$
surrogates suitable for spherical raindrops.
Note that Eqs. (15)-(17) are derived for the range over which the data of WJ00 is valid (e.g., 1 <
Re < 120), and they should not be used for extrapolation to Re values larger or smaller than this
range. Since Re < 1 corresponds to ice particle smaller than $D_{thres}$, it is justified to assume that $E_c$ =
0 in this Re range. When considering the range Re > 120, values of $E_c$ for Re = 120 should be
used; this is reasonable based on the experiments of K74 for 200 < Re <640, and the theoretical
study of WJ00 for 60 $\leq$ Re $\leq$ 120.



**5.2  Hexagonal columns**
H80 and M95 did not provide any $E_c$ equation for columnar crystals. To the best of our knowledge,
there is not any practical equation for such crystals in the literature, suitable for use in cloud
resolving models. In addition to hexagonal plates, WJ00 studied $E_c$ between hexagonal columns
(with width $w$ between 47 and 292.8 μm, length $l$ between 67.1 and 2440 μm and $0.2 < \text{Re} < 20$)
and water drops of $1\ \mu m < d < 100\ \mu m$. Note that WJ00 calculated Re for columns in a different
way than was done for plates. Re for columns was calculated from their width, whereas Re for
plates was computed from their maximum dimension (e.g., $\text{Re}_{columns} = wV/\varepsilon$ ). If the values of Re
were calculated from the column maximum dimension, they would have values comparable to
those for plates. In formulating $E_c$ for columns, we have followed the Re convention of WJ00.
Similar to hexagonal plates, we provide the best fits to the data of WJ00 for hexagonal columns:

$$E_c = \begin{cases} \left(0.787K^{0.988}\right)\left(-0.0121\text{Re}^2 + 0.1297\text{Re} + 0.0598\right), & 0.01 \le K \le K_{thres} \ \& \ 0.2 \le \text{Re} \le 3 \\[6pt] \left(0.787K^{0.988}\right)\left(-0.0005\text{Re}^2 + 0.1028\text{Re} + 0.0359\right), & 0.01 \le K \le K_{thres} \ \& \ 3 < \text{Re} \le 20 \\[10pt] r\sqrt{1 - \dfrac{1}{3.5}\left[\log(\dfrac{K}{K_{crit}}) - \sqrt{3.5}\right]^2}, & K_{thres} < K < 20 \ \& \ 0.2 \le \text{Re} \le 20 \end{cases} \quad (18)$$

where $K$ is calculated from Eq. (16), and $K_{\text{crit}}$ is calculated as:

$$K_{crit} = \begin{cases} 0.7779\,\text{Re}^{-0.009}, & 0.2 \le \text{Re} \le 1.7 \\ 1.0916\,\text{Re}^{-0.635}, & 1.7 < \text{Re} \le 20 \end{cases} \quad (19)$$

and $r$ is a parameter related to the major radius of the ellipse fit and is determined as:

$$r = \begin{cases} 0.8025\,\text{Re}^{0.0604}, & 0.2 \le \text{Re} \le 1.7 \\ 0.7422\,\text{Re}^{0.2111}, & 1.7 < \text{Re} \le 20 \end{cases} \quad (20)$$

and $K_{\text{thres}}$ is calculated as:

$$K_{thres} = \begin{cases} 0.0251\text{Re}^2 - 0.0144\text{Re} + 0.811, & 0.2 \le \text{Re} \le 2 \\ -0.0003\text{Re}^3 + 0.0124\text{Re}^2 - 0.1634\text{Re} + 1.0075, & 2 < \text{Re} \le 20 \end{cases} \quad (21)$$



The results are shown in Fig. 5b. Similar to hexagonal plates, the curve fits are able to represent
the data of WJ00 very well over various ranges of $K$ and Re. The percent error in $E_c$ between the
curve fits and the WJ00 data has a mean value of 10.28% with a standard deviation of 5.81% for
all Re and $K$. There are no experimental estimates of $E_c$ for hexagonal columns in the literature for
comparison. For a given $K$, $E_c$ of columnar ice crystals increases with increasing in Re (due to the
increase in fallspeed). For a given Re, $E_c$ increases with increasing in $K$ (because of increasing
droplet diameter), but it does not exceed 0.95. Unlike plates, the increase in Re does not decrease
the sensitivity of $E_c$ to Re.
Again, Eqs. (18)-(21) should not be used for Re < 0.2 and Re > 20. In the range Re < 0.2, the
column size does not exceed the $D_{\text{thres}}$, and therefore $E_c = 0$. For Re > 20, values of $E_c$ are
unknown, but we suggest using $E_c$ for Re = 20 as a conservative estimate of $E_c$.

## 6  Mass growth rate by riming

In Sect. 4, the dependence of $\alpha$ on IWC was explained. Unrimed IWC can be derived from $\alpha$ and $\beta$
pertaining to unrimed ice crystals (see EM16). Rimed IWC can be calculated by using the
definition of riming mass growth rate, similar to Heymsfield (1982), M95 and JH15:

$$\left(\frac{dm}{dt}\right)_{ri\min g} = \int_0^{d_{\max}} A_g(D,d)|V(D)-v(d)|E(D,d)m(d)n(d)dd \tag{22}$$

where $t$ is time, $A_g(D,d)$ is the geometrical cross-section area of the ice particle-cloud droplet
collection kernel, $E(D,d)$ is collection efficiency between the cloud droplet and ice particle, $m(d)$ is
the cloud droplet mass, $n(d)$ is the cloud droplet number density, and $d_{\max}$ is diameter of the largest
cloud droplet. Note that the cloud droplet sedimentation velocity $v(d)$ is negligible compared to the
ice particle fallspeed $V(D)$ and is assumed to be zero in the similar equation by Heymsfield (1982),
M95, and Zhang et al. (2014). Zhang et al. (2014) used a different equation, which has the form of
$dm/dt = A(D)V(D)E(D)LWC$, where LWC is equal to $\int_0^{d_{\max}} m(d)n(d)dd$. For this equation, the
riming rate is not sensitive to the droplet distribution.





Based on the observations of Locatelli and Hobbs (1974), many cloud and climate models use a $V$-
$D$ power law to predict ice mass sedimentation rates ($V = a_v D^{b_v}$, with constant $a_v$ and $b_v$ for each
specific particle habit; Rutledge and Hobbs, 1984; Ferrier, 1994; Fowler et al., 1996; Pinski et al.,
1998; Morrison and Gettelman, 2008; Gettelman and Morrison, 2015). However, such a
relationship cannot represent the evolution of ice particle size and shape, and is often inconsistent
with the realistic dependence of $V$ on the ice particle $m/A$ ratio. This increases uncertainty in the
microphysical and optical properties of such models. To overcome this issue, M96 introduced a
method that derives $V$ by using $m$ and $A$, and also by a power law for the Best number ($X$) and Re
relationship ($\mathrm{Re} = AX^B$, where $A$ and $B$ are constant coefficients in specific ranges of $X$). In this
method, the $V$ calculation depends on the $m/A$ ratio. Mitchell and Heymsfield (2005) followed the
same method, but they used a Re-$X$ power law with variable coefficients ($A$ and $B$ are not constant
anymore) to produce a smooth transition between different flow regimes. Such an approach is
shown to represent the evolution of $V$ realistically (MG08; Morrison and Grabowski, 2010; JH15;
Morrison and Milbrandt, 2015).
Since the contribution of the cloud droplet projected area to $A_g(D,d)$ is negligible, $A_g(D,d)$ can be
approximated as the maximum ice particle cross-section area projected normal to the air flow. Ice
particles fall with their major axis perpendicular to the fall direction, therefore $A_g(D,d)$ is
approximated as the ice particle $A$, which is calculated in Sect. 4.2. The $m(d)$ is calculated from
spherical geometry as: $m(d) = \pi d^3 \rho_w / 6$. $E(D,d)$ is equal to $E_c E_s$ where $E_c$ was discussed in Sect.
5, and $E_s$ is the sticking efficiency (fraction of the water droplets that stick to the ice particle after
collision), and is presumed to be unity since supercooled cloud droplets freeze and bond to an ice
particle upon collision. Conditions under which $E_s$ may be less than unity are addressed in
Pruppacher and Klett (1997). It is noteworthy that by using the above calculations, riming growth
will be represented in a self-consistent, gradual, and continuous way. Based on the explanations in
this section, Eq. (22) can be reduced to:

$$\left(\frac{dm}{dt}\right)_{ri\min g} = A(D)V(D)\int_0^{d_{\max}} E(D,d)m(d)n(d)dd .$$

(23)





Differentiating Eq. (1) with respect to $t$ corresponds to $dm/dt = D^{\beta}\,d\alpha/dt + \alpha\beta D^{\beta-1}dD/dt$, but
the second term on the RHS should be relatively small (riming has little impact on $D$ prior to
graupel formation). Therefore, to a first approximation:

$$\left(\frac{d\alpha}{dt}\right)_{riming} = \frac{1}{D^{\beta}}\left(\frac{dm}{dt}\right)_{riming}, \qquad\qquad (24)$$

and together with Eq. (23), a change in $\alpha$ due to riming can be determined.
Figure 6 shows $dm/dt$ calculated from Eq. (23) for hexagonal ice plates for different values of
LWC and droplet median-mass diameter (MMD; the droplet diameter that divides the droplet PSD
mass into equal parts). $E_c$ is calculated from Eq. (15), and a sub-exponential PSD is assumed for
cloud droplets that has the form:

$$n(d) = N_o d^{\nu}\exp(-\lambda d), \qquad\qquad (25)$$

where $\lambda$ is the PSD slope parameter, $\nu$ is the PSD dispersion parameter and $N_o$ is intercept
parameter. M95 used observational droplet spectra from Storm Peak lab (Steamboat, Colorado,
USA), and calculated various PSD parameters: $\nu = 9$, $\lambda = (\nu+1)/\overline{d}$, and $N_o = 4\times10^{4}\,LWC/\rho_w\overline{d}^{13}$
, where $\overline{d}$ is droplet mean diameter, and is related to MMD as $MMD = 1.26\overline{d}$ for this dataset.
Note that all variables are in units of cgs. It is seen in Fig. 6 that $dm/dt$ increases with increasing
ice particle $D$. The $dm/dt$ is linearly proportional to LWC when MMD and $D$ are constant. In
addition, when LWC is constant, doubling MMD (from 8 to 16 μm) leads to a quadrupling of
$dm/dt$. One important feature is the contribution of small droplets ($d < 10$ μm) to $dm/dt$, when $K <$
0.7 and $E_c < 0.3$. It is seen in this figure that when MMD is relatively small (= 8 μm), ignoring
such small droplets results in values of $dm/dt$ at the largest crystal sizes that are ~ 0.25% of those
obtained when all droplets are included. This is due to half of the LWC being associated with $d < 8$
μm. However, when MMD is larger (= 16 μm), the effect of small droplets is only ~ 5%. The
collection kernel ($K_c$) can be calculated as $A(D)V(D)E(D,d)$, which is alternatively equal to $dm/dt$
divided by LWC (see Eq. 23). MG08 approximated this variable by using simple assumptions, and
found that it is proportional to $D^2$. Here, we showed by more accurate analysis that $K_c$ has a form



of second-order polynomial fit, and is represented by $K_c = 7 \times 10^{-6} D^2 - 0.0002D + 0.0008$ for
MMD = 8 μm.
## 7   Conclusions
In most atmospheric models, riming is treated as an abrupt change between precipitation classes;
from snow to graupel, which occurs at an arbitrary threshold size. Such parameterizations are not
realistic and lead to uncertainty in the simulation of snowfall. In this study, a combination of
various empirical and theoretical approaches is utilized to shed light on the riming process. SCPP
ground-based measurements of $m$ and $D$ for rimed and unrimed ice particles are used in this study;
such particles represent ice clouds for -40 °C < $T$ < 0 °C. The findings presented here suggest a
fundamental shift in our way of representing ice particle $m$ and $A$ in atmospheric models for
riming. It is common in most models to assume that riming increases $\beta$ (Eq. 1) from values of ~ 2
(for dendrites) to values of ~ 3 (for graupel). However, we showed that this assumption is not
supported by observations. To a good approximation under most conditions, riming does not
increase (or decrease) $\beta$ and $D$ in an $m$-$D$ power law and the treatment of riming is simplified with
riming increasing only $\alpha$. To represent unrimed particles in frontal clouds, one could enlist the
polynomial fit for synoptic ice clouds (-40°C < $T$ < -20°C, see EM16) but adjust this equation to
conform to the observed power laws for unrimed dendrites. To treat riming for dendrites, this fit
equation could be multiplied by the riming fraction $m_r/m_u$ or alternatively IWC/IWC$_u$. A similar
strategy could be adopted for other ice particle shapes or shape mixtures in frontal clouds, as is
done for columnar particles in this study. By using this method, there is no discontinuity in the
growth of $m$ and $A$; rather, the particles grow gradually during riming process.
There is no practical method to calculate $E_c$ in models for columnar crystals. Moreover, most
models use the H80 equation to calculate $E_c$ for planar crystals, but this equation has important
drawbacks inherited from the early numerical studies (See Sect. 1.2). To solve this problem, new
equations for the calculation of $E_c$ are developed based on the numerical study of WJ00 for both
hexagonal plates and hexagonal columns that accounts for dependence of $E_c$ on cloud droplet $d$
and ice particle $D$ in non-steady flow. In the future, this treatment of the riming process will be





employed in a new SGM that predicts the vertical evolution of ice particle size spectra in terms of
the growth processes of vapor diffusion, aggregation and riming.
**Acknowledgements**
This research was supported by the Office of Science (BER), U.S. Department of Energy. We are
grateful to Brad Baker for providing us with the measurements of ice particle projected area that
were used in BL06. The SCPP data used in this study and associated software is freely available to
interested researchers; those interested should contact the second author.

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



**Figure Captions**
Figure 1. (a) Comparing the *m-D* curve fit based on the CPI and cold-habit SCPP data (EM16)
with SCPP ice particle *m-D* measurements corresponding to all classifiable shapes. Unrimed and
rimed particles are indicated by blue and red dots, respectively. *m-D* power laws from two other
studies are also displayed. (b) Similar to (a), except that all the SCPP data (including unclassifiable
ice particles) have been grouped into size-bins; mean (red cross-intersection points) and standard
deviation (red bars) in each size-bin are shown.
Figure 2. Ice particle *m-D* measurements corresponding to rimed (pink dots) and unrimed (blue
dots) dendrites using SCPP data. Mean (circles) and standard deviations (bars) in each size bin are
also displayed for both rimed (red) and unrimed (black) dendrites. Green filled circles indicate
dendrites from BL06.
Figure 3. Rimed-to-unrimed mass ratio $m_r/m_u$ (violet lines) for each common size-bin in Figure 2,
based on heavily rimed and unrimed dendrites. The pink line indicates the weighted mean of
$m_r/m_u$. The numbers on the top (bottom) of each violet line shows the number of rimed (unrimed)
particles in that size bin.
Figure 4. (a) Same as Fig. 3, but rimed particles are now graupel. (b) Same as (a), but unrimed
particles are now columnar crystals and R4a (hexagonal graupel) is not included.
Figure. 5. (a) Collision efficiency as a function of mixed Froude number. Circles show the data of
WJ00 based on numerical calculations, and curves show the best fits to this data for various values
of Re. Also displayed are experimental data of ST73 for Re = 97 (squares), K74 for $200 \leq Re \leq$
640 (diamonds), and K74 for $10 \leq Re \leq 35$ (triangles). (b) Same as (a), but for hexagonal columns
and no experimental data.
Figure. 6. Riming mass growth rate versus hexagonal plate *D* for various LWC (0.05, 0.1 and 0.2 g
m$^{-3}$) and different droplet median-mass diameters (8 and 16 μm). Additional curves (dotted dashed
and dotted curves) are produced by assuming that $E_c$ conforms to the ellipse curves and therefore is
zero for smaller droplets ($d < 10$ μm).





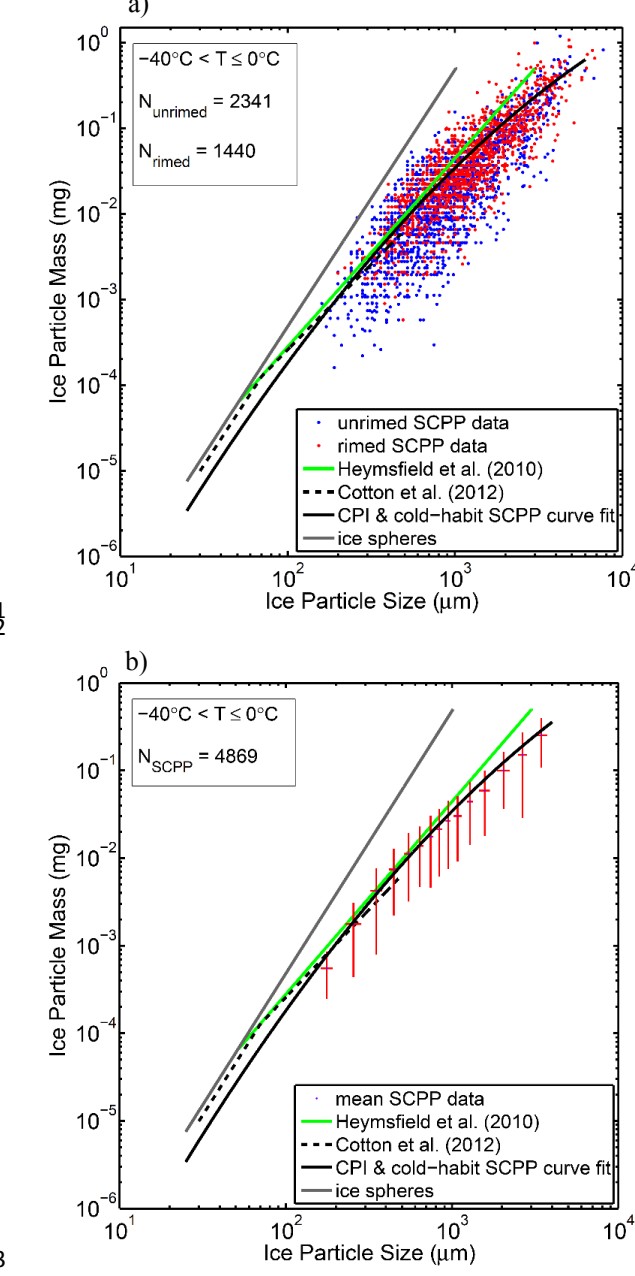

Figure 1. (a) Comparing the *m-D* curve fit based on the CPI and cold-habit SCPP data (EM16) with SCPP
ice particle *m-D* measurements corresponding to all classifiable shapes. Unrimed and rimed particles are
indicated by blue and red dots, respectively. *m-D* power laws from two other studies are also displayed. (b)
Similar to (a), except that all the SCPP data (including unclassifiable ice particles) have been grouped into
size-bins; mean (red cross-intersection points) and standard deviation (red bars) in each size-bin are shown.



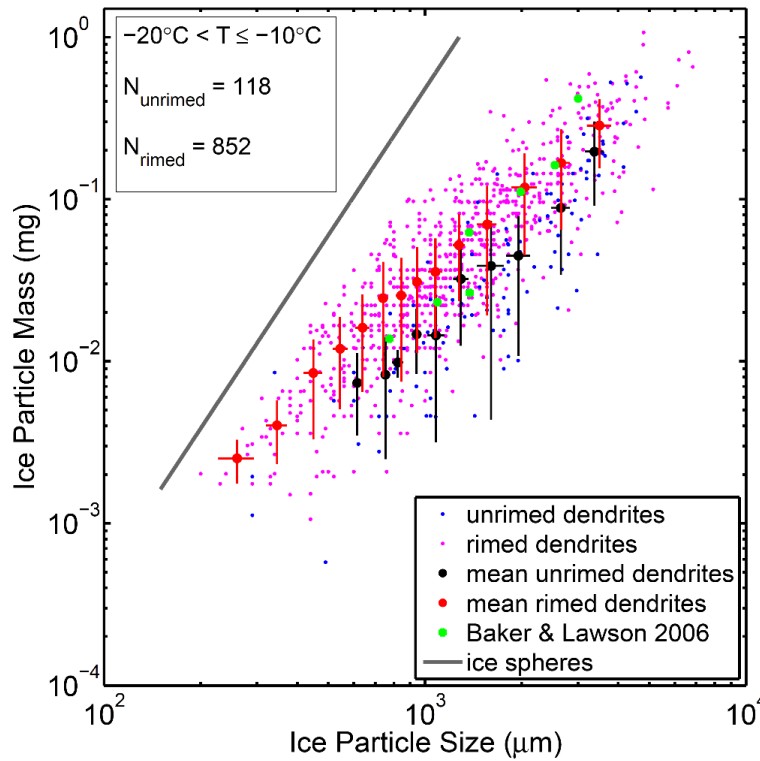

3    Figure 2. Ice particle *m-D* measurements corresponding to rimed (pink dots) and unrimed (blue dots)
4    dendrites using SCPP data. Mean (circles) and standard deviations (bars) in each size bin are also displayed
5    for both rimed (red) and unrimed (black) dendrites. Green filled circles indicate dendrites from BL06.





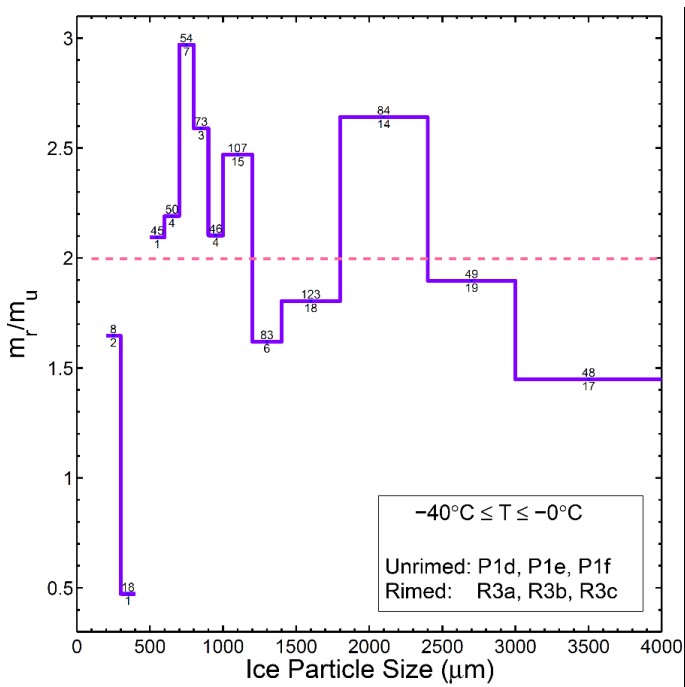

Figure 3. Rimed-to-unrimed mass ratio $m_r/m_u$ (violet lines) for each common size-bin in Figure 2, based on

heavily rimed and unrimed dendrites. The pink line indicates the weighted mean of $m_r/m_u$. The numbers on

the top (bottom) of each violet line shows the number of rimed (unrimed) particles in that size bin.





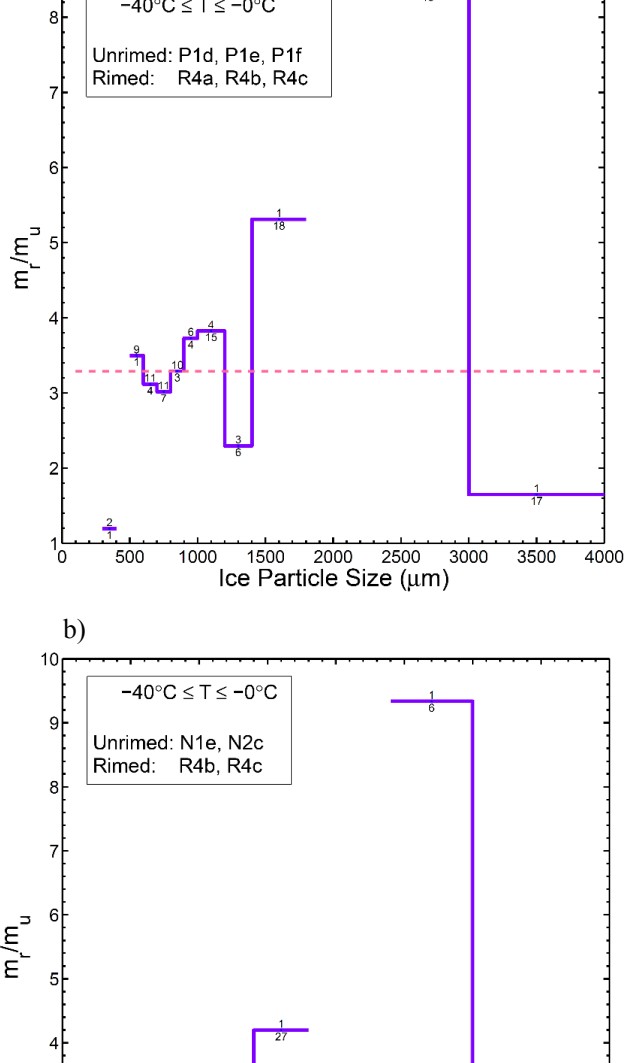

3  Figure 4. (a) Same as Fig. 3, but rimed particles are now graupel. (b) Same as (a), but unrimed particles are

4  now columnar crystals and R4a (hexagonal graupel) is not included.





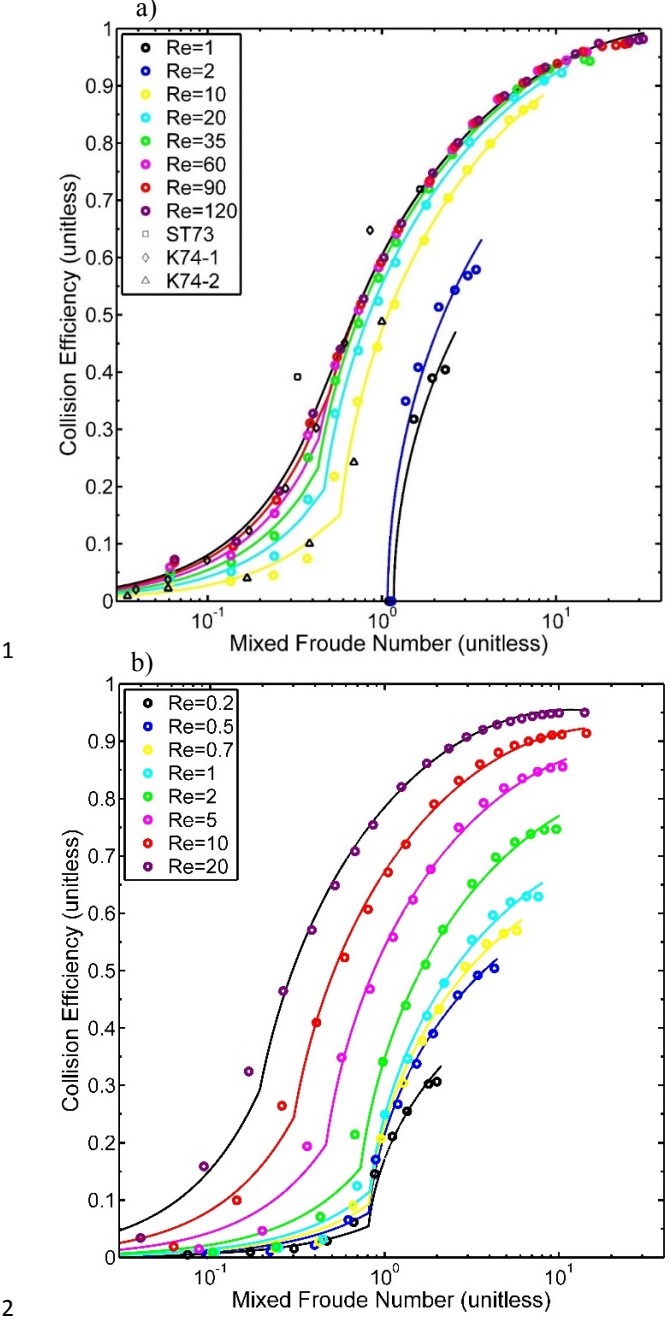

Figure. 5. (a) Collision efficiency as a function of mixed Froude number. Circles show the data of WJ00 based on numerical calculations, and curves show the best fits to this data for various values of Re. Also displayed are experimental data of ST73 for Re = 97 (squares), K74 for $200 \leq Re \leq 640$ (diamonds), and K74 for $10 \leq Re \leq 35$ (triangles). (b) Same as (a), but for hexagonal columns and no experimental data.



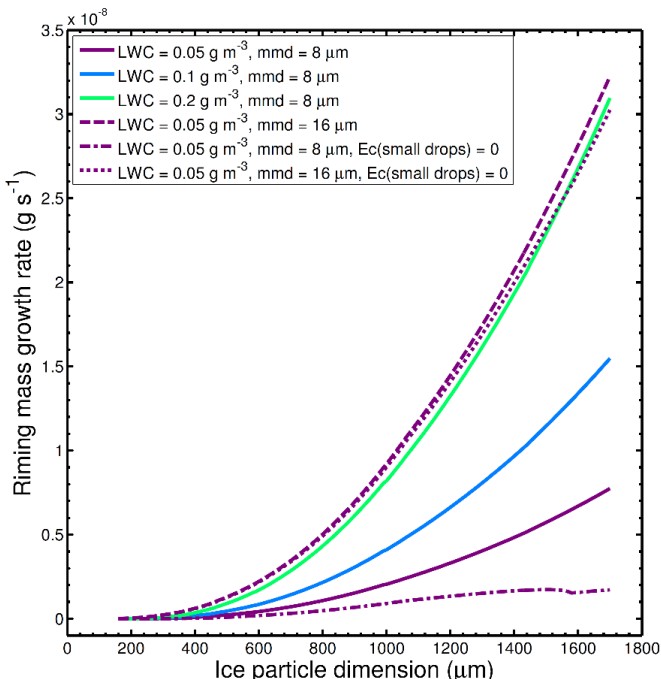

Figure. 6. Riming mass growth rate versus hexagonal plate *D* for various LWC (0.05, 0.1 and 0.2 g m$^{-3}$) and
different droplet median-mass diameters (8 and 16 µm). Additional curves (dotted dashed and dotted
curves) are produced by assuming that $E_c$ conforms to the ellipse curves and therefore is zero for smaller
droplets ($d < 10$ µm).

