# Peer review of "Growth of ice particle mass and projected area during riming Ehsan Erfani1,2,\* and David L. Mitchell1 [1] {Desert Research Institute, Reno, Nevada, USA} [2] {Graduate Program in Atmospheric Sciences, University of Nevada, Reno, Nevada, USA} [\*] {Now at Center for Ocean-Land-A"

_Atmospheric Chemistry and Physics, 2016_

## Referee Comment (RC1) · Anonymous Referee #1 · 4 Jul 2016

Review of "Growth of ice particle mass and projected area during riming" From Ehsan Erfani and David L. Mitchell

Overview: The authors investigate the riming effects in frontal cloud systems in the sierra Nevada, on coefficients of mass-size and area-size relationships. They used a dataset (referenced as SCPP in the paper) described in Mitchell et al. 1990 and also used in Mitchell (1996) and Baker and Lawson (2006) who are often cited in recent scientific studies. In the third section, SCPP dataset is compared to a recent mass-diameter fitted curve. This fitted m-d relation was developed with mass and diameter derived from CPI images using the area-mass relationships given in Baker and Lawson 2006, and a part of the SCPP dataset (call cold habit). CPI images represent ice particles for temperature colder than -20°C in cirrus and anvil clouds during the aircraft measurement of the field campaign SPARTICUS. Ice particle of the SCPP dataset are

divided in two parts: rimed and unrimed particles. Authors shows that rimed particles are heavier than unrimed particles for a same size. Furthermore, ice particles of the SCPP dataset are stored by size bins where average mass are calculated and compared to the EM16 mass-diameter relationship, authors conclude that the EM16 fitted curve is representative of mass-size relationships of ice particles in clouds for temperatures between -40°C and -20°C. Section 4 is dedicated to the effects of riming on mass-size and area-size coefficients. In a first step the study is performed with heavily rimed and unrimed dendrites where mass-size relations are calculated for each type of dendrites. It shown that heavily rimed dendrites have around twice time the mass of unrimed dendrites in average and that power of m-d relations are similar for both types of dendrites: 1.78. Hence, authors found that riming process affects only $\alpha$ in m-d power law and that increase of $\alpha$ is a proxy to study the evolution of the riming process. Then a riming factors is defined as the ratio the mass of the ice crystal less its mass when it is unrimed divided by the difference between the mass of the ice crystal when it is a graupel and its mass when it is unrimed. Then weighted average ratio are calculated between unrimed dendrites and graupel and between hexagonal columns and graupel. Calculated ratios are: 3.3 and 2.4 respectively, which define the maximum mass of dendrites habit and columns habit such mmax=3.3*mu (u for mass of unrimed crytals) and mmax=2.4*mu, respectively. In section 5 authors provide fits of collision efficiency between ice crystals such (columns and plates) and supercooled droplets using data of WJ00. The work is done for different Froude number and Stokes number. Others references are used to avoid discontinuities in different regimes. Section 6 is dedicated to the calculation of the mass growth rate due to the riming process. To do this, results of section 5 are used. The method is applied for hexagonal plates and a droplet size distributions following a Gamma function. Results obtained show that dm/dt for riming increase with D of plates. Furthermore, for a same Liquid water content doubling mmd from 8 to 16 microns leads to times by 4 dm/dt. Note that a LWC of 0.05 g/m3 with mmd of 16 microns gives dm/dt of the same order than for a LWC of 2g/m3 and a mmd of 8microns. Increase mmd is more efficient than increase LWC

General comments: The topic of this study could be very useful for experimental and numerical simulations studies about physical processes of ice hydrometeors in clouds. However, I find that this work is not finalized and it could be improved with further investigations. Also it does not meet the basic criteria of ACP: does the study "represent a substantial contribution to scientific progress within the scope of Atmospheric Chemistry and Physics (substantial new concepts, ideas, methods, or data)" Authors shows only results for one type of ice crystals habits that the power of m-d stay constant during the riming. This result was already suggested in former studies which are cited in this study. Some results of the M90 dataset argue against the main results of this paper without being discussed. Also the M90 dataset is not appropriate to extend this kind of results for in-clouds habits conclusions, as it concerns only precipitated ice particles. Results should be take more cerfully. Section on area-size relationships should be strongly reconsidered as it based on no new experiments or numerical simulation data, there is only a discussion about older results which brings no new results. I recommend to rewrite this paper by keeping section five and developing section 6 for not only hexagonal plates but also hexagonal columns and dendrites habits (has these habits are used in this paper) while section 4 could be used to compare numerical results of section 6. Mass growth rate could be compared to the mass of unrimed ice crystals as a function of D, to confirm or not the experimental results who shows that power of m-D relation is constant during the riming process until the graupel formation. Moreover, it could be extended to a theoretical/numerical study on the riming factors as a function of time which could be useful for future studies with ground radar observations versus aircraft measurement. To finish this study could compare numerical results with experimental data as SPARTICUS and SCPP datasets.

Specific comments: P8, l3-l5: what kind of modification, is it possible to resume with few words? P9, l2: As the riming is the purpose of the paper. I suggest a figure with few examples of microprobe images to show each riming level of ice crystals who are studied. P9, l3: is it the ground temperature, or temperature are deduced using diagram (for example Pruppacher and Klet 1997; Magono and Lee 1966; Bailey and

Hallet 2009), then few explanations are expected. P9, l6 (P10, l10): CPI images doesn't give the mass of Ice crystals. Which approximation did you use to deduce the mass of ice crystals from CPI images? P11, l6-l11: Is there rimed particles in the cold habit and SPARTICUS fit? Or do you know if the ratio of rimed/unrimed particles are the same for both datasets in the comparison? If the ratio of unrimed and rimed particles are not equivalent, does it means that the fit curve is representative of ice clouds between -40°C and -20°C. "To summarize, it appears . . .. A realistic bulk estimates for ice particle masses in frontal clouds." A part of SCPP dataset is used to fit the curve of cold habit? Then what is the ratio of the SCPP dataset compared to SPARTICUS dataset in this curve fit? Because the conclusion is not surprising if the rate of SCPP dataset is significant compared to SPARTICUS dataset. P13, l3-l4: Watching the same table1 in M90, I can see that it is not true for needles and rimed needles. So it seems that it is not true for all ice crystals habit. P14, l4-l15: there is missing in your statistical analysis, the both dataset that you conclude are identical should have an equivalent ratio of rimed and unrimed ice particles. Especially if you want generalize your conclusions to all frontal clouds. P15-P16: Definition of mmax and $\alpha$max. mmax define the mass of one ice crystal when it can be considered as a graupel, as $\beta$ is constant during the riming so $\alpha$ increase until m=mmax. After what you define a riming factor such R=(m-mu)/(mmax-mu). By looking the figure below, we can guess that mmax ($\alpha$max) is a function of D, as R. Maybe more investigations should be necessary (and/or maybe an application on the SPARTICUS dataset, or another dataset) to use R factors and understand how it quantify riming in clouds. P17, l16-l22: I think that BL06 study is also performed on the SCPP dataset!?

Technical corrections: Missing references for: P3, l6: Feng and Chang (1982) P4, l7: Fukuta and Takahashi (1999) P8, l6: Blohn et al. (2009) P9, l17: Magono and Lee (1966) P17, l8-l9: Baker and Lawson (2006) P23, l1-l14: a reference to Heymsfield and Westbrook 2010 is recommended. They also worked on the terminal velocities of ice crystals and improved Mitchell 1996 theory. Figures: Figure1a: mathematical Formula could be a plus. A curve fit for rimed and unrimed particle should be added. Figure2:

I recommend a plot of curves fit of rimed, unrimed dendrites and graupel. Figure 1 and 2: green curve and points are not visible enough. P6, l9: ice particle shapes are function of temperature and relative humidity (see Pruppacher and Klett 1997; Magono and Lee 1966; Bailey and Hallet 2009) P3, l20: "Mf = IWC times Vm"? P3, l14: NWS is not defined.

- ice sphere
- hail M96
- graupel M96
- graupel Erfani and Mitchell
- dendrites unrimed
- dendrites unrimed (D<3000μm)
- dendrites rimed
- 3.3*dendrites unrimed
- 3.3*dendrites unrimed (D<3000μm)

Axis labels: $m$ [g] (vertical), $D_{max}$ [cm] (horizontal)

**Fig. 1.**

---

## Referee Comment (RC2) · Anonymous Referee #2 · 2 Aug 2016

This paper provides a way to evolve m-D and A-D relationships during riming with the goal of being able to improve riming in models that rely on these m-D and A-D parameterizations for different ice categories. The authors use their own previous data fit to compare with observations from the Sierra Cooperative Pilot Project. They show that riming increases the mass of ice and argue that this increase in mass can be parameterized without changing the beta coefficient in the m-D relationships. They also provide collection efficiency parameterizations for both planar and columnar ice based on numerical calculations. While the goal of this paper is to improve models by using observations, this paper has major issues that need addressed before it can be published. Major issues: Perhaps be more specific about how to use this method to improve models that rely on different ice categories. It seemed hard to follow just how and when rimed snow would become spherical or hexagonal graupel using this

framework.

By assuming graupel forms when the mass of ice has increased approximately three fold, assumptions about the type of graupel and it's density are being made and this should be clarified.

Perhaps the introduction needs an extra section on the treatment of riming in models after talking about the characteristics of riming. Also, perhaps the riming rate equation needs to be introduced (at least qualitatively) in the introduction before talking about riming and collection efficiencies.

m-D relationships are parameterizations which reduce the dimensionality of complex ice shapes. More discussion is needed on alpha and beta and what they mean. Alpha can be considered to contain density and thickness information, thus certainly it can contain all the change to m-D during riming, but others could argue that because shape changes during riming beta should change at some point. The current ideas in this paper and comparison to observations are useful because they link theory with modeling and confirm the idea that light and moderate riming can be modeled with assuming fixed beta, and this method should improve models. But where does, for example, the fixed beta assumption break down? What are the limitations of the method?

General comments. Why not show a plot of alpha evolving or d alpha/dt? Page 3 Line 4 "mass of snow collected" Lines 7-8 perhaps change "snowfall rate" to "precipitation rate" Line 10 40% (extra space) Line 13 (QPEs) Line 14 remove (SGM) as it is only used one other time and not often Line 18 change to "mass sink" Line 21 change word "powerful" General comment: watch the spacing between number and deg C or other units Line 29 what is considered the western Arctic? Page 4 Lines 1-2 are confusing, reword Line 3 remove "tracks", make storms on the previous line plural Line 6 this doesn't limit understanding of riming, instead it limits understanding of phase partitioning Line 9 riming peaks at -10.5C for low LWCs. Also, the reason is partly due to different collection efficiences for planar versus isometric ice Line 13 remove "process" Line 14 "with a spherical shape" Lines 14-16 both sentences start with "In this". Change Line 23 give a better definition of alpha and beta Page 5 Line 6 make clear that these are all different for different ice types Line 20 autoconversion isn't an actual word, so maybe just use conversion Line 20 "hydrometeor category to another" Line 24 "abrupt microphysical changes" to what? Line 28 This isn't necessarily 4-moment because rime fraction and rime volume are not moments of the size distribution. You may want to ask Dr. Morrison about this. Line 29 there is no need to note about aggregation as it is not discussed in this paper Page 6 Lines 1-2 you may want to better describe the method of MG09 Line 23 put Re in parentheses Line 24 what is a free fallspeed Line 25 "that were exposed to" Line 26 "Although d ranged" Page 8 Line 21 "from the Sierra" Line 29 and following line both start with Also Page 9 Line 15 remove hyphens after moderately and heavily Line 17 add a year to the Magono and Lee citation Page 10 Line 3 the binning intervals description is confusing. Line 7 Do the results change if lightly rimed ice is assumed to be in the rimed ice category? Figure 1 what density is assumed for the solid ice spheres? Lines 14-15 This is shown in Figure 2 and this should be noted Page 11 Line 8 20-30% is less than other get and is less than the value of about a doubling of mass you use later in the paper. How can you explain this? Line 15 perhaps start of by talking about methods that assume constant beta (MG09) and then corroborate their results Page 12 Eq 3 Can you also plot best fit values assuming $D^3$? What reduced density values would you get if you assumed $D^3$ and is this reasonable? Lines 9-10 Put these statements in better context. Perhaps talk about how bulk models use graupel categories ($D^3$) and some assume spherical snow, but other assume snow goes as $D^2$. Then talk about how beta should change in nature and the best way to parameterizing riming snow in models. Line 16 Again, if you assumed spherical graupel, what density would it have and could you also plot that m-D curve? Page 13 Line 5 "beta as constant during the riming process (until spherical) for both" Line 8 perhaps use "number distribution" instead of "number density" Line 9 remove "because it is a function only of D" Lines 12-18 seem out of place Line 20 this value is inconsistent with the 20-30% from earlier. Explain. Page 14 Lines 1-3 you can

probably just say this is due to errors in classification Page 15 Line 10 Is this also justified because beta is approximately 2? Line 19 The value of 3.3 will depend on things like rime density. Under what conditions does using this value work? Page 17 Line 5 again the value of 2.4 will depend on a lot of variable. When do you expect the model to break down? Page 22 Line 5 doesn't increasing size also lead to increasing Re? Line 11 "as a conservative underestimate" Line 19 perhaps "number distribution" Line 21 it is assumed to be negligible, not zero Page 23 Line 7 "microphysical and therefore optical" Page 24 Line 11 is it really $d^{13}$? Page 25 Line 11 should this be "representing ice particle m and D"? Page 26 Line 1 change SGM to simple growth model

---

## Author Comment (AC1) · 11 Dec 2016

Dear editor,

Attached please find our response and our track-changed manuscript in the form of a supplement.

Best regards, Authors

Please also note the supplement to this comment:
http://www.atmos-chem-phys-discuss.net/acp-2016-455/acp-2016-455-AC1-supplement.pdf

---

## Author Response (AR1)

**Response to the referee comments on the manuscript:**

Title: Growth of ice particle mass and projected area during riming
By: Erfani, Ehsan; Mitchell, David
Article reference: acp-2016-455

We wish to thank the referees for their detailed and helpful comments on our paper. As you will see below we have responded to all of the comments with revisions designed to address the concerns of the referees. In the following response, the original referee comments appear in black and our responses appear in blue and are labeled "Author response:"

**Referee comments:**

**Anonymous Referee #1:**

Review of "Growth of ice particle mass and projected area during riming" From Ehsan Erfani and David L. Mitchell

Overview:

The authors investigate the riming effects in frontal cloud systems in the sierra Nevada, on coefficients of mass-size and area-size relationships. They used a dataset (referenced as SCPP in the paper) described in Mitchell et al. 1990 and also used in Mitchell (1996) and Baker and Lawson (2006) who are often cited in recent scientific studies. In the third section, SCPP dataset is compared to a recent mass-diameter fitted curve. This fitted m-d relation was developed with mass and diameter derived from CPI images using the area-mass relationships given in Baker and Lawson 2006, and a part of the SCPP dataset (call cold habit). CPI images represent ice particles for temperature colder than -20°C in cirrus and anvil clouds during the aircraft measurement of the field campaign SPARTICUS. Ice particle of the SCPP dataset are divided in two parts: rimed and unrimed particles. Authors shows that rimed particles are heavier than unrimed particles for a same size. Furthermore, ice particles of the SCPP dataset are stored by size bins where average mass are calculated and compared to the EM16 mass-diameter relationship, authors conclude that the EM16 fitted curve is representative of mass-size relationships of ice particles in clouds for temperatures between -40°C and -20°C. Section 4 is dedicated to the effects of riming on mass-size and area-size coefficients. In a first step the study is performed with heavily rimed and unrimed dendrites where mass-size relations are calculated for each type of dendrites. It shown that heavily rimed dendrites have around twice time the mass of unrimed dendrites in average and that power of m-d relations are similar for both types of dendrites: 1.78. Hence, authors found that riming process affects only $\alpha$ in m-d power law and that increase of $\alpha$ is a proxy to study the evolution of the riming process. Then a riming factors is defined as the ratio the mass of the ice crystal less its mass when it is unrimed divided by the difference between the mass of the ice crystal when it is a graupel and its mass when it is unrimed. Then weighted average ratio are calculated between unrimed dendrites and graupel and between hexagonal columns and graupel. Calculated ratios are: 3.3 and 2.4 respectively, which define the maximum mass of dendrites habit and columns habit such mmax=3.3*mu (u for mass

of unrimed crystals) and mmax=2.4*mu, respectively. In section 5 authors provide fits of collision efficiency between ice crystals such (columns and plates) and supercooled droplets using data of WJ00. The work is done for different Froude number and Stokes number. Others references are used to avoid discontinuities in different regimes. Section 6 is dedicated to the calculation of the mass growth rate due to the riming process. To do this, results of section 5 are used. The method is applied for hexagonal plates and a droplet size distributions following a Gamma function. Results obtained show that dm/dt for riming increase with D of plates. Furthermore, for a same Liquid water content doubling mmd from 8 to 16 microns leads to times by 4 dm/dt. Note that a LWC of 0.05 g/m3 with mmd of 16 microns gives dm/dt of the same order than for a LWC of 2g/m3 and a mmd of 8microns. Increase mmd is more efficient than increase LWC.

General comments: The topic of this study could be very useful for experimental and numerical simulations studies about physical processes of ice hydrometeors in clouds. However, I find that this work is not finalized and it could be improved with further investigations. Also it does not meet the basic criteria of ACP: does the study "represent a substantial contribution to scientific progress within the scope of Atmospheric Chemistry and Physics (substantial new concepts, ideas, methods, or data)"

Author response: While it is not clear why this reviewer feels that this work is not finalized and not a "substantial contribution", it should be mentioned that this is the 2[nd] paper in a three-part series that began with an extensive development of ice particle m-D and A-D relationships (now published in ACP; Erfani and Mitchell, 2016). The 3[rd] paper (soon to be submitted to ACPD) presents a new snow growth model (SGM) that predicts the steady-state height evolution of the ice particle size distribution (PSD) due to ice particle growth by vapor diffusion, riming and aggregation (as well as updraft effects on the PSD evolution). The riming treatment in this SGM is based on this 2[nd] paper, and the SGM is tested against the evolution of observed PSD in a deep mixed phase cloud. These three papers certainly constitute a "substantial contribution", but the first ACP article alone is 22 pages, and to combine all these papers into a single article does not make sense as per normal journalistic standards. Combining the 2[nd] and 3[rd] papers would also result in an article of unacceptable length.

Another reason for limiting this paper to the development and description of a scheme for modeling the riming process is to present this scheme in a manner that will be general for use in most bulk microphysical parameterizations (BMPs). Most BMPs use m-D and A-D power laws to parameterize the microphysical processes, but no BMPs that we know of have represented these power laws as a function of riming. This representation is needed in BMPs to realistically treat the riming process, and this paper provides a means of doing this. Thus, we argue that a compelling need exists to improve BMPs in terms of the riming process, and that providing a general approach to address this need is indeed a "substantial contribution".

Authors shows only results for one type of ice crystals habits that the power of m-d stay constant during the riming. This result was already suggested in former studies which are cited in this study.

Author response: In addition to show the results for dendrites, we showed this for all ice particles in SCPP data with various shapes. Regarding the originality of our work: first we are aware of only one observational study (Rogers, 1974) that suggested the conservation of $\beta$ for snowflakes

during the riming process. No other observational study investigated this result. Second, the modeling study of Morrison and Grabowski (2008; henceforth MG08), Morrison and Grabowski (2010) and Morrison and Milbrandt (2015) used the result of Rogers (1974) as a basis for their models. In addition, Morrison and Milbrandt (2015) also used Table 1 in Mitchell et al. 1990 (henceforth M90) for hexagonal columns and our preliminary results (from our conference paper: Mitchell and Erfani, 2014) for dendrites to support the assumption of constant $\beta$ for their model. Only a very limited number of models utilized this method (constant $\beta$ during phase 1 of riming), and we think that further investigation of this assumption by extra observations will be very useful for atmospheric modeling. Thus, our study is not a repetitive study. Finally, investigation of $\beta$ during the riming was only a part of our study. One important achievement in our study is developing a method for growth of projected area, which is unprecedented. Previous studies were focused on mass and not projected area. In fact, MG08 mentioned the discontinuity in projected area that they observed in their model. Another important result of our study is the formulation of the cloud droplet-ice crystal collision efficiency for planar and columnar ice crystals, based on more recent numerical results from fluid-dynamic modeling studies. Such calculations were needed because the vast majority of cloud models use a collision efficiency formulation from an old study that is based on an older theoretical data with many limitations and assumptions. Therefore, we think that our numerically practical treatment of the riming process may be very significant in the field of atmospheric science and can be easily used in climate and cloud models.

Some results of the M90 dataset argue against the main results of this paper without being discussed. Also the M90 dataset is not appropriate to extend this kind of results for in-clouds habits conclusions, as it concerns only precipitated ice particles. Results should be take more carefully.

Author response: We would be happy to discuss any discrepancy between our results and M90 if such disagreement was mentioned explicitly.

M90 dataset is a unique available dataset that measures both ice particle size and mass, as well as the degree of riming, for many ice particle shapes. Aircraft do not provide such measurements, and there is no other dataset that we know of having these attributes and that suits our purpose. Cloud base was not far from the sample collection site, and sublimation effects were not evident based on crystal shapes. As explained in Erfani & Mitchell (2016, ACP; henceforth EM16), the important problem with airborne measurements is that they are unable to measure single ice particle mass. Nonetheless, we compare our results with m-D relationships from previous studies (Heymsfield et al., 2010; Cotton et al., 2012; EM16) that are based on airborne measurements.

We added this explanation to Data and Method, page 9, starting at line 5:

"SCPP is a unique dataset that measures both ice particle size and mass and also determines the degree of riming. As explained in EM16, the important problem with airborne measurements is that they are unable to measure single ice particle mass. Nonetheless, we compare our results with m-D relationships from previous studies (Heymsfield et al., 2010; Cotton et al., 2012; EM16) that are based on airborne measurements."

Section on area-size relationships should be strongly reconsidered as it based on no new experiments or numerical simulation data, there is only a discussion about older results which brings no new results. I recommend to rewrite this paper by keeping section five and developing section 6 for not only hexagonal plates but also hexagonal columns and dendrites habits (has

these habits are used in this paper) while section 4 could be used to compare numerical results of section 6. Mass growth rate could be compared to the mass of unrimed ice crystals as a function of D, to confirm or not the experimental results who shows that power of m-D relation is constant during the riming process until the graupel formation.

Author response: The use of a measurement database in other studies does not disqualify its use in new studies if the new study has found a unique purpose for the database. It is possible that this SCPP database has been used in other studies due to its uniqueness (these measurements are extremely labor-intensive) and the utility of these single-particle measurements. Our study has found a new application for this SCPP database; the treatment of m-D and A-D relationships as a function of the riming process as described in Sect. 4.

Although Sect. 4.2. on the effect of riming on projected area has some limitations (and we mentioned the limitations at the beginning of this section), we think this section is a new theoretical approach that is important for these reasons: MG08 used different *A-D* power laws in each riming step, but such method led to discontinuities in projected area during the transition from one ice category to another one. It seems that the *A-D* and *m-D* that they used were not self-consistent (e.g. they were from different studies based on different datasets). We understand this challenge to find self-consistent m-D and A-D expressions, and as we mentioned, there are no projected area-dimensional measurements in SCPP data. Therefore, we conclude that by calculating area from mass (as described by Eqns. 10 & 11) there will be no discontinuity in area. We used the results of previous studies (such as Fontaine et al., 2014; Heymsfield 1978; Jensen & Harrington, 2015; henceforth JH15) to develop our riming formulation. Our formulation is a new development, and it is not a mere repackaging of previous findings. In addition, we use this approach in our future paper (Erfani et al., 2017) and show how it improves the simulation of projected area during the riming process. We added this explanation at the beginning of Sect. 4.2:

"MG08 used different *A-D* power laws in each riming step, but this method led to discontinuities in projected area during the transition from one ice category to another one. It seems that the *A-D* and *m-D* that they used were not self-consistent (e.g. they were from different studies based on different datasets). Here, we suggest an approach to avoid the discontinuity in projected area."

Regarding the development of Sect. 6, we added an extra panel in Fig. 6 to show the mass growth rate for hexagonal columns (Fig. 6.b), along with explanations in the text. It is not feasible for us to extend the research for dendrites, because it strongly depends on the derivation of collision efficiency for dendrites, which has not been done for this study and it needs extra time and resources which is not possible for us to do at this time. The explanations for hexagonal columns mass growth rate have been added to the manuscript at the end of the second-to-last paragraph in Sect.6:

"One important feature is the contribution of small droplets ($d < 10$ μm) to *dm/dt*, when $K < 0.7$ and $E_c < 0.3$. It is seen in this figure that when MMD is relatively small (= 8 μm), ignoring such small droplets results in values of $(dm/dt)_{riming}$ at the largest crystal sizes that are ~ 40% (for plates) and ~ 70% (for columns) of those obtained when all droplets are included. That is, small droplets contribute about 60% and 30%, respectively, to the $(dm/dt)_{riming}$ values at the largest sizes. This surprising contribution from small droplets is partly due to half of the LWC being associated with $d < 8$ μm. However, when MMD is larger (= 16 μm), the contribution from small droplets is only ~ 5%. The size-dependence of *dm/dt* for hexagonal columns (Fig. 6b) shows that

*dm/dt* for columns is larger than that for hexagonal plates for a specific crystal size when droplet MMD is 8 μm, partly because columns fall faster than plates (see Fig. 6 in M96) and partly due to higher $E_c$ for columns encountering larger droplets. Moreover, when LWC is constant, doubling MMD (from 8 to 16 μm) leads to at least a doubling of *dm/dt* (greater for plates)."

Regarding the calculation of mass growth rate for unrimed ice crystals, it needs to calculate mass growth rate for microphysical processes other than riming (e.g. vapor deposition and aggregation), which is beyond the scope of this study. However, our snow growth model (SGM), which is the subject our future paper (Erfani et al., 2017) includes three growth processes of vapor deposition, aggregation, and riming, and your question in this regard will be answered in our future paper. This is reflected at the end of conclusions, page 25, starting at line 28:

"… In the future, this treatment of the riming process will be employed in a new SGM that predicts the vertical evolution of ice particle size spectra, mass, projected area, fallspeed, and snowfall rate in terms of the growth processes of vapor diffusion, aggregation and riming. These results will be compared with airborne measurements from two spiral descents."

Moreover, it could be extended to a theoretical/numerical study on the riming factors as a function of time which could be useful for future studies with ground radar observations versus aircraft measurement. To finish this study could compare numerical results with experimental data as SPARTICUS and SCPP datasets.

Author response: This is the subject of another paper that we are preparing (Erfani et al., 2017). We developed a snow growth model (SGM) to simulate microphysical processes of vapor deposition, aggregation and riming. In addition, aircraft data from two Lagrangian spiral descents are used to compare the model with observations.

Specific comments:

P8, lines 3-5: what kind of modification, is it possible to resume with few words?

Author response: We added these explanations in this part of manuscript for clarification:

"Similar to Beard and Grover (1974), they employed the superposition method for collision between particles, but they assumed that the small cloud droplets do not change the graupel fallspeed. Therefore they used the Stokes number instead of the mixed Froude number in the non-dimensionalized momentum equation (see Eqs. 1-6 in Rasmussen and Heymsfield, 1985)."

P9, line 2: As the riming is the purpose of the paper. I suggest a figure with few examples of microprobe images to show each riming level of ice crystals who are studied.

Author response: Since the high-magnification photos taken during SCPP do not contain examples of rimed ice crystals corresponding to these 4 riming levels, we cite the Magono and Lee ice particle classification scheme that was used to classify ice particle shape during SCPP. The 4 riming levels used come from the Magono and Lee scheme, which provides illustrations for each ice particle type. The following text was added: "Also recorded were individual ice particle shapes, which were classified using the Magono and Lee (1966) nomenclature scheme. The level of riming (i.e. light, moderate, heavy riming, or graupel) was indicated based on this scheme, and the temperature range over which the observed ice particle shape originated was recorded (e.g. for long columns, -8°C < T < -6°C). These riming levels are indicated (with rimed

crystal illustrations) in the Magono and Lee (1966) ice particle classification scheme with the prefix R1, R2, R3 and R4 (see Pruppacher and Klett, 1997, p. 46). Photographic examples of these rimed particle types are shown in Fig. 2 of Locatelli and Hobbs (1974)."

P9, line 3: is it the ground temperature, or temperature are deduced using diagram (for example Pruppacher and Klet 1997; Magono and Lee 1966; Bailey and Hallet 2009), then few explanations are expected.

Author response: The latter is used. This has been clarified in the above paragraph that was added to the manuscript.

P9, line 6 (P10, line 10): CPI images doesn't give the mass of Ice crystals. Which approximation did you use to deduce the mass of ice crystals from CPI images?

Author response:

Appendix B in Erfani and Mitchell (2016) explained the method to approximate CPI mass. In the first paragraph of this Appendix B it is written: "There is no direct measurement of ice particle mass by the CPI probe. Moreover, the Baker and Lawson (2006) m-A relationship is based on ice particles larger than 150 μm. Therefore, we developed a new method for estimating mass based on CPI measurements of ice particle projected area, length and width. It is assumed that when 10 μm < D < 100 μm, all ice crystals are hexagonal columns. The apparent aspect ratio, defined as the CPI measured mean length-to-width ratio for a given size interval, is generally between 1 and 2 in this size range and the ice crystals are known to be relatively dense (more mass per maximum dimension), making this shape assumption a reasonable approximation (Korolev and Isaac, 2003; Lawson et al., 2006). This is more accurate than assuming ice particles to be spherical."

A brief explanation has been added to this part of the manuscript:

"Since CPI does not measure ice particle mass, EM16 developed a method that calculates mass from the measurements of ice particle projected area, maximum dimension, and aspect ratio by assuming that small ice particles can be approximated as hexagonal columns (for more details, see Appendix B in EM16)."

P 11, lines 6-11: Is there rimed particles in the cold habit and SPARTICUS fit? Or do you know if the ratio of rimed/unrimed particles are the same for both datasets in the comparison? If the ratio of unrimed and rimed particles are not equivalent, does it means that the fit curve is representative of ice clouds between 40°C and -20°C. "To summarize, it appears .... A realistic bulk estimates for ice particle masses in frontal clouds." A part of SCPP dataset is used to fit the curve of cold habit? Then what is the ratio of the SCPP dataset compared to SPARTICUS dataset in this curve fit? Because the conclusion is not surprising if the rate of SCPP dataset is significant compared to SPARTICUS dataset.

Author response: This section was not well written and apparently produced some confusion. The writing of this section has been improved, especially the first sentence: "The purpose of this section is to demonstrate how the cold habit SCPP curve fit from EM16 (based on unrimed ice crystals) compares with all the SCPP data, since this shows how the EM16 curve fit appears representative for all ice particles sampled during SCPP and thus may be representative for Sierra Nevada snowfall."

As stated in Sec. 2, there are 827 ice particles in the cold habit (EM16) SCPP dataset and there are 4869 ice particles in the total SCPP dataset.

P13, lines 3-4: Watching the same table1 in M90, I can see that it is not true for needles and rimed needles. So it seems that it is not true for all ice crystals habit.

Author response: Yes, while this is true, only 7 rimed needles were sampled in M90 ($\beta = 2.1$) and only 16 unrimed needles were sampled ($\beta = 1.8$; same as for unrimed and rimed columns). Thus, the statistics are too poor to say much about $\beta$ for rimed needles. We mentioned rimed and unrimed hex columns since these have relatively good statistics and are graphically displayed and analyzed in Sec. 4d of M90, along with aggregates of rimed and unrimed hex columns (which also support our evidence of constant $\beta$ during riming).

P14, lines 4-15: there is missing in your statistical analysis, the both dataset that you conclude are identical should have an equivalent ratio of rimed and unrimed ice particles. Especially if you want generalize your conclusions to all frontal clouds.

Author response: We used the same dataset (SCPP) that contains both rimed and unrimed particles. In order to be realistic, we did not change the ratio of rimed to unrimed ice particles, and instead we used the measurements with the observed ratio.

P15-P16: Definition of mmax and $\alpha$max. mmax define the mass of one ice crystal when it can be considered as a graupel, as $\beta$ is constant during the riming so $\alpha$ increase until m=mmax. After what you define a riming factor such R=(m-mu)/(mmax-mu). By looking the figure below, we can guess that mmax ($\alpha$max) is a function of D, as R. Maybe more investigations should be necessary (and/or maybe an application on the SPARTICUS dataset, or another dataset) to use R factors and understand how it quantify riming in clouds.

Author response: Riming does not occur for broad-branched crystals with D less than 200 μm (Wang and Ji, 2000; this is also mentioned in the paper). Therefore, the extrapolation of the curves to D < 200 μm is not physically-meaningful. In addition, the data size range should be taken into consideration. The mean-bin size in SCPP data does not exceed 3500 μm, and any extrapolation larger than this size should be taken with caution. In fact, we showed in the paper that the uncertainty for the largest size bin is higher for unrimed dendrites, and it might be safe to use the second largest size bin (D = 2800 μm) as the upper range.

Based on Figs. 3 and 4 in our manuscript, we do not see any meaningful relationship between the mass ratio and *D* in SCPP data. There is no riming information in SPARTICUS data, so we cannot use it for this purpose. Therefore we cannot make any conclusion or guess about the dependence of mass ratio on D. Indeed, Figs. 3 and 4 suggest that there is no such dependency.

P17, lines 16-22: I think that BL06 study is also performed on the SCPP dataset!?

Author response: As explained in Erfani and Mitchell (2016), BL06 used a subset of ice particles (865 particles) measured during SCPP. Using image analysis software, the projected area of ice particles in this subset was calculated from their photographed magnified images. The BL2006 study found that ice particle projected area was a more reliable predictor of particle mass than was maximum dimension. Their m-A power law was derived from many types of ice particle habits or shapes; of the 550 identifiable ice particles, 36% were moderately or heavily rimed.

This m-A power law is now commonly used to estimate size-resolved mass concentrations from 2D-S probe measurements of projected area. Since BL06 used only a subset of the SCPP data to produce a *m-A* relationship (not a *m-D* relationship), comparison of their work and our study is meaningful.

This explanation has been added to this paragraph:

"BL06 used a subset of SCPP data (e.g. 865 ice particles), of which 550 were identifiable, and 36% of such identifiable particles were moderately or heavily rimed. They then developed a software to calculate ice particle projected area from their magnified images. Thereafter, they calculated a *m-A* power law expression. Since BL06 used only a subset of the SCPP data to produce a *m-A* relationship (i.e. not a *m-D* relationship), comparison of their work and our study is meaningful."

Technical corrections: Missing references for: P3, l6: Feng and Chang (1982) P4, l7: Fukuta and Takahashi (1999) P8, l6: Blohn et al. (2009) P9, l17: Magono and Lee (1966) P17, l8-l9: Baker and Lawson (2006)

Author response: All the missing references have been added to the list. There is no Feng and Chang (1982) in the manuscript, instead the reference for Feng and Grant (1982) is added. The reference for von Blohn et al. (2009) has been in the reference list (number 3 from the end).

P23, line 1-14: a reference to Heymsfield and Westbrook 2010 is recommended. They also worked on the terminal velocities of ice crystals and improved Mitchell 1996 theory.

Author response: We added this reference and explanations to this part of manuscript. Now, the manuscript reads (starting at P23, L14, in original manuscript):

"In addition, Heymsfield and Westbrook (2010) developed an alternative method to improve M96 method, and calculated $X$ as a function of $m/A_r$ ratio, where $A_r$ is area ratio (defined as the ratio of ice particle projected area to the projected area of a circumscribed circle around the particle; see Eq. 15 in Erfani and Mitchell, 2016)."

Figures: Figure1a: mathematical Formula could be a plus. A curve fit for rimed and unrimed particle should be added.

Author response: The mathematical formula for polynomial curve fit of CPI and cold-habit SCPP is added to this figure. The main message of this figure is to show how the SCPP data conforms to various m-D expressions from literature. The rimed and unrimed curve fits overlap with the current m-D expressions, and prevent to convey the message of this figure. Instead, we added the rimed and unrimed curve fits to Fig. 2

Figure2: I recommend a plot of curves fit of rimed, unrimed dendrites and graupel.

Author response: The m-D power laws for unrimed and rimed dendrites and graupel have been added to this figure.

Figure 1 and 2: green curve and points are not visible enough.

Author response: We understand that green might not be as sharp as black, blue, and red, but we already used these colors as points or curve. If we use colors other than green or yellow, it will be very similar to blue, red, or their combination, and it will not be easy to distinguish between them and the red, blue or black curves or points. In addition, we will provide figures with higher resolution for publication, and that should decrease the problem.

P6, line 9: ice particle shapes are function of temperature and relative humidity (see Pruppacher and Klett 1997; Magono and Lee 1966; Bailey and Hallet 2009)

Author response: This sentence has been changed and the relative humidity dependency has been added.

P3, line 20: "Mf = IWC times Vm"?

Author response: The multiplication sign has been added: "$M_f = \text{IWC} \times V_m$"

P3, line 14: NWS is not defined.

Author response: The definition of NWS (National Weather Service) has been added.

**Anonymous Referee #2:**

This paper provides a way to evolve m-D and A-D relationships during riming with the goal of being able to improve riming in models that rely on these m-D and A-D parameterizations for different ice categories. The authors use their own previous data fit to compare with observations from the Sierra Cooperative Pilot Project. They show that riming increases the mass of ice and argue that this increase in mass can be parameterized without changing the beta coefficient in the m-D relationships. They also provide collection efficiency parameterizations for both planar and columnar ice based on numerical calculations. While the goal of this paper is to improve models by using observations, this paper has major issues that need addressed before it can be published.

Major issues:

Perhaps be more specific about how to use this method to improve models that rely on different ice categories. It seemed hard to follow just how and when rimed snow would become spherical or hexagonal graupel using this framework. By assuming graupel forms when the mass of ice has increased approximately three fold, assumptions about the type of graupel and its density are being made and this should be clarified.

Author response:

In order for models with multiple ice categories to use our method, they should add two transitional phases during riming growth. Phase 1 simulates the growth of an ice crystal from unrimed crystal up to the graupel onset. In this phase, mass and projected area gradually increase, but size is unchanged. Phase 2 represents graupel growth. In this phase, the shape is unchanged, but mass, projected area, and size gradually increase.

We used lump graupel and cone-like graupel (R4b and R4c) from SCPP dataset to determine the rimed-to-unrimed mass ratio for hexagonal columns. These same graupel shapes with the addition of hexagonal graupel (R4a) are used to calculate this ratio for hexagonal plates. These are explained in Sect. 4. Since there is no information on particle density in SCPP data, we did not make any assumption on the graupel density. However, SCPP observations show the conservation of D and β during the phase 1 of riming, and since the initial graupel mass is ~3 times larger than unrimed crystal mass, these mean that initial graupel density is ~ 3 times larger than unrimed crystals. Ice particle density generally implies spherical geometry, making the concept somewhat artificial for non-spherical ice particles. Nonetheless, one can easily calculate the density from the m-D relationship, defining spherical volume with D.

The explanations on the application of our method to models have been added as a new paragraph to Conclusions, page 25, starting at line 23:

"It is straightforward for models with multiple ice categories to utilize our new method. This can be done by describing riming growth as two phases and removing the autoconversion process. Phase 1 simulates the growth from ice crystal to the onset of graupel formation. In this phase, mass and projected area gradually increase, but size is unchanged (Eqs. 6-11). Phase 2 represents

graupel growth. In this phase, the shape is unchanged, but mass, projected area, and size gradually increase (Eqs. 12-14)."

And the explanation on the assumptions for graupel shape and density is added to Sect. 4.2.1, page 15, starting at line 14:

"In order to determine $m_{\max}$, we calculated the $m_r/m_u$ that corresponds to graupel (R4a, R4b, and R4c) and unrimed dendrites (P1d, P1e, and P1f), as shown in Fig. 4a. Small variability is seen for $D$ < 1200 μm (ranges from 3 to 3.8, with the exception of smallest size bin), whereas large variability exists (between 1.6 and 8.4) for larger sizes due to the small number of graupel in each size bin. The weighted average for this $m_r/m_u$ ratio is equal to 3.3 which can be used to estimate $m_{\max}$: $m_{\max} \approx 3.3 \times m_u$ for the dendrites. Since R4a occurs just before hexagonal features are completely obscured by additional rime deposits, R4a graupel is ideal for estimating $m_{\max}$. Unfortunately, there are only 14 R4a particles in the entire SCPP data, with $D$ < 1200 μm. They exhibit a large variability in the $m_r/m_u$ ratio (ranging from 1.6 to 4.5) with a weighted average of $m_r/m_u$ equal to 3.1 (figure not shown). Nonetheless the close agreement with the above $m_r/m_u$ ratio of 3.3 is encouraging for us to conclude that initial graupel mass (at the end of phase 1) is 3.3 times larger than unrimed dendrites. Since the SCPP observations show that $D$ and $\beta$ are conserved during the first phase of riming, graupel density is also ~ 3.3 times larger than unrimed dendrite mass."

Perhaps the introduction needs an extra section on the treatment of riming in models after talking about the characteristics of riming. Also, perhaps the riming rate equation needs to be introduced (at least qualitatively) in the introduction before talking about riming and collection efficiencies.

Author response: We added an extra section (now: 1.2. treatment of riming in models) in the Introduction after the characteristics of riming and before the collision efficiency. We moved a paragraph on modeling of the riming (previously at the end of Sect. 1.1) to this section. We also added more explanations on the method of MG08. Moreover, an explanation on riming mass growth rate is added to this new section:

"**1.2. Treatment of Riming in Models**

Since explicit modeling of the riming process is computationally expensive, graupel and hail categories were not considered in some bulk microphysics parameterizations used in some global climate models or GCMs (Morrison and Gettelman, 2008; Gettelman and Morrison, 2015). The common ice microphysics approach in most cloud and climate models is the separation of ice into various hydrometeor categories such as cloud ice, snowflakes, and graupel (Rutledge and Hobbs, 1984; Ferrier, 1994; Fowler et al., 1996; Reisin et al. 1996; Morrison and Gettelman, 2008; Gettelman and Morrison, 2015). The transition between various hydrometeors occurs by autoconversion from one hydrometeor category to another. However, such autoconversion is arbitrary and poorly constrained, and as shown by Eidhammer et al. (2014), cloud radiative properties were sensitive to the choice of autoconversion threshold size in the Community Atmosphere Model version 5 (CAM5). This is because the distinct boundaries between various ice hydrometeor categories impose abrupt changes in microphysical properties (such as ice

particle mass, projected area, fallspeed, and effective diameter) from one hydrometeor category to another, while in nature the transition processes are gradual.

To overcome this problem, MG08 advanced a bulk model that employed vapor diffusion and the riming processes, and used multiple *m-D* and *A-D* power laws (Eqs. 1 and 2) to characterize ice particles associated with different parts of the PSD. This method was applied to a bin model developed by Morrison and Grabowski (2010), and was later used in a bulk model that prognoses number concentration, rime volume, rime mass and total mass (Morrison and Milbrandt, 2015). In this approach, different *m-D* and *A-D* expressions from the literature are used for either pure ice crystals or graupel. Then, *m* and *A* are calculated in a transitional step as a function of *D* and rimed mass fraction. Such *m-D* and *A-D* expressions resulted in a gradual transition from crystal mass to graupel mass. However, discontinuities were observed in transition between various *A-D* expressions over the PSD because the size range for each microphysical step (ice crystal, graupel, and transition) was calculated in a way to provide only continuous mass, and thus produced discontinuities in projected area. JH15 developed a detailed ice growth model that simulates ice particle habit and mass via the processes of vapor deposition and riming. This model is also a single-category scheme, but it does not employ *m-D* and *A-D* power laws; instead, it computes the growth of ice particles along the major and minor axes of oblate or prolate spheroids (representing hexagonal plates or columns). Therefore, the model is able to simulate simple ice particle shapes, and also captures the temperature-dependency of vapor deposition and the riming processes (since particle shape is a function of temperature and relative humidity; Magono and Lee 1966; Pruppacher and Klett 1997). The simulated ice particle shape, mass, and fallspeed are in good agreement with observational data from wind tunnel experiments on ice crystal growth.

The computation of rime mass (an increase in ice particle mass by riming) in models is performed by calculating the accretional mass growth rate (Heymsfield, 1982; Mitchell, 1995; JH15). When an ice particle falls in a cloud of supercooled cloud droplets, the increase in its mass due to accretion depends on ice particle features (e.g. fallspeed and projected area), droplet characteristics (e.g. mass and number concentration of droplets), and the collision efficiency ($E_c$) between an ice particle and droplet. More details on mass growth rates are provided in Sect. 6, and $E_c$ is described in the next section."

m-D relationships are parameterizations which reduce the dimensionality of complex ice shapes. More discussion is needed on alpha and beta and what they mean. Alpha can be considered to contain density and thickness information, thus certainly it can contain all the change to m-D during riming, but others could argue that because shape changes during riming beta should change at some point.

Author response:

We added extra sentences to explain the importance of m-D relationships and to define α and β. Erfani and Mitchell (2016) explained m-D and A-D relationships with more details. Regarding the change in β, we explain that with more details in the next comment. β starts to change at the beginning of phase 2 of riming, where ice particle has become graupel and has a quasi-spherical

shape. The explanations on m-D expressions are added to manuscript, page 4, starting from line 21:

"Many models calculated ice particle mass by assuming that ice particles are spherical (e.g. $m = \rho_i \pi D^3/6$, where $\rho_i$ is ice density; Rutledge and Hobbs, 1984; Ferrier, 1994; Morrison and Gettelman, 2008). However, this assumption is not realistic, and produces errors in the evolution of snow-size spectra (Mitchell, 1988). Based on observations, several studies developed ice particle mass-dimension (*m-D*) power law parameterizations to reduce the dimensionality of complex ice particle shapes. For a specific ice particle shape or an environmental condition, such relationship has the form:

$$m = \alpha D^\beta,$$ (1)

where both $\alpha$ and $\beta$ are constants over a specific size range. They are determined via direct measurements of ice particle mass and dimension (Locatelli and Hobbs, 1974; M90), or are constrained through aircraft measurements of the ice particle size distribution (PSD) and IWC (Heymsfield et al., 2010; Cotton et al., 2012). The prefactor $\alpha$ was considered to contain information on particle density and thickness, whereas $\beta$ was believed to have information on particle shape. We will discuss the latter in Sect. 4.1 for the riming process."

The current ideas in this paper and comparison to observations are useful because they link theory with modeling and confirm the idea that light and moderate riming can be modeled with assuming fixed beta, and this method should improve models. But where does, for example, the fixed beta assumption break down? What are the limitations of the method?

Author response:

Your explanations are right and refer to phase 1 of riming growth in our paper. During phase 1, the shape of ice particles changes, but the maximum dimension D does not. That is, both the mass and projected area increase during phase 1 in a way that mass and area increase by rime-deposits filling in the gaps between branches in the case of dendrites, or making the column more spherical by expanding the a-axis. The assumption of conserved β and δ appear valid in phase 1. The ice crystal growing by riming turns into a graupel particle at the end of phase 1. Once the graupel threshold is reached, phase 2 growth commences that describes the growth of quasi-spherical graupel. In this phase, D does increase due to riming, and so do mass and projected area. In particular, phase 1 is demonstrated by Eqs. (6)-(11), whereas phase 2 is explained by Eqs. (12)-(14). To better clarify and distinguish between these two phases, we added some explanations to the manuscript that explicitly mention phase 1 and 2:

abstract:

"… It was observed that $\beta$ in the *m-D* power law $m = \alpha D^\beta$ appears independent of riming during the phase 1 (before the formation of graupel), with $\alpha$ accounting for the ice particle mass increase due to riming. … Once the graupel with quasi-spherical shape forms, $D$ increases with an increase in $m$ and $A$ (phase 2 of riming). …"

Page 16, starting from line 10:

"So far, we discussed the phase 1 of riming growth (before the formation of graupel), where $m$ and $A$ increases while $D$ and therefore $\beta$ and $\delta$ are conserved. Once the graupel stage is attained, phase 2 of riming starts and the graupel continues to grow through riming, and a different methodology is required to describe riming growth at this growth stage, because graupel $D$ increases by riming. …"

Conclusions, first paragraph, page 25, starting from line14:

".… To a good approximation under most conditions, riming does not increase (or decrease) $\beta$ and $D$ in an $m$-$D$ power law and the treatment of riming is simplified with riming increasing only $\alpha$ during the phase 1 of riming (before the formation of graupel). …"

Page 25, end of the first paragraph, starting from line 22:

"… Phase 2 of riming starts when graupel with quasi-spherical shape forms. In this phase, the increase in $m$ and $A$ causes an increase in $D$."

General comments.

Why not show a plot of alpha evolving or d alpha/dt?

Author response: $d\alpha/dt$ is calculated in Eq. (24) and is implied from Fig. 6. Since D and $\beta$ do not change by riming, $d\alpha/dt$ is linearly proportional to dm/dt. We will show the change in $\alpha$ with more details in our future paper (Erfani et al., 2017).

We added this sentence to Sect. 6, page 24, starting at line 4:

"… Since $D$ and $\beta$ do not change by riming, $d\alpha/dt$ is linearly proportional to $dm/dt$."

Page 3 Line 4 "mass of snow collected"

Author response: Corrected.

 Lines 7-8 perhaps change "snowfall rate" to "precipitation rate"

Author response: We feel it is better not to change this since "snowfall rate" is used earlier in this paragraph, and the meaning is clear.

 Line 10 40% (extra space)

Author response: Corrected.

 Line 13 (QPEs)

Author response: Corrected

Line 14 remove (SGM) as it is only used one other time and not often

Author response: Done

Line 18 change to "mass sink"

Author response: We added the mass sink in this part:

"... should be affected by the mass sink, represented by the ice mass flux ..."

Line 21 change word "powerful"

Author response: We changed it to considerable

General comment: watch the spacing between number and deg C or other units

Author response: We checked all the units to make sure that there is a space between the number and unit, based on the ACP writing style.

Line 29 what is considered the western Arctic?

Author response: It refers to the Northwest Territories, Canada. This synonym is added to the sentence:

"... they happen nearly half of the time in the western Arctic, also known as Northwestern Territories, Canada (Shupe et al., 2006) ..."

Page 4 Lines 1-2 are confusing, reword

Author response: Changed to:

"... Moreover, such clouds are an important part of tropical convective storms, as airborne observations indicate large amounts of supercooled water in these storms (Rosenfeld and Woodley, 2000). ..."

Line 3 remove "tracks", make storms on the previous line plural

Author response: Changed.

Line 6 this doesn't limit understanding of riming, instead it limits understanding of phase partitioning

Author response: Changed.

Line 9 riming peaks at -10.5C for low LWCs. Also, the reason is partly due to different collection efficiences for planar versus isometric ice

Author response: We added this explanation to this sentence:

"They also showed that riming has a peak at -10.5 °C for low LWCs, where ice particles are isometric, and therefore have higher $V$. In addition, the reason is partly due to different collision efficiencies for planar versus isometric ice particles."

Line 13 remove "process"

Author response: Changed

Line 14 "with a spherical shape"

Author response: Changed.

Lines 14-16 both sentences start with "In this". Change

Author response: For the second one, "In this way" is changed to "Therefore".

Line 23 give a better definition of alpha and beta

Author response: Since Equation (1) makes it obvious that $\alpha$ is the prefactor and $\beta$ is the power exponent, now we simply state: "where both $\alpha$ and $\beta$ are constants over a specific size range."

It is not straightforward to define $\alpha$ and $\beta$ from a physical point of view, and this text is part of the Introduction. Most papers just say alpha and beta are constants. Nonetheless, we added this explanation to clarify these constants (page 5, starting at line 11):

"$\alpha$ was considered to contain information on particle density and thickness, whereas $\beta$ was believed to have information on particle shape. We will discuss the latter in Sect. 4.1 for the riming process."

Page 5 Line 6 make clear that these are all different for different ice types

Author response: We clarified that this happens for a single crystal habit:

"For a specific ice habit, the $m$-$D$ and $A$-$D$ power laws are dependent on the size range considered, and it often takes two or even three $m$-$D$ power laws to describe a given $m$-$D$ relationship over all relevant sizes for that habit."

Line 20 autoconversion isn't an actual word, so maybe just use conversion

Author response: "autoconversion" is now very common in the field of cloud microphysics modeling to describe the change from one ice category to another one in a model. Conversion is a general word that does not convey our purpose.

Line 20 "hydrometeor category to another"

Author response: Changed.

Line 24 "abrupt microphysical changes" to what?

Author response: To address this comment, we changed this sentence:

"…This is because the distinct boundaries between various ice hydrometeor categories impose abrupt changes in microphysical properties (such as ice particle mass, projected area, fallspeed, and effective diameter) from one hydrometeor category to another, while in nature the transition processes are gradual. …"

Line 28 This isn't necessarily 4-moment because rime fraction and rime volume are not moments of the size distribution. You may want to ask Dr. Morrison about this.

Author response: We deleted "4-moment" from this sentence:

"… and was later used in a bulk model that prognoses number concentration, rime volume, rime mass and total mass. …"

Line 29 there is no need to note about aggregation as it is not discussed in this paper

Author response: We deleted the mention of aggregation.

Page 6 Lines 1-2 you may want to better describe the method of MG09

Author response: We added new explanations to describe the method of MG08, in page 6, starting from line 1:

"…In this approach, different $m$-$D$ and $A$-$D$ expressions from the literature are used for either pure ice crystals or graupel. Then, $m$ and $A$ are calculated in a transitional step as a function of $D$ and rimed mass fraction. Such $m$-$D$ and $A$-$D$ expressions resulted in a gradual transition from crystal mass to graupel mass. However, discontinuities were observed in transition between various $A$-$D$ expressions over the PSD because the size range for each microphysical step (ice crystal, graupel, and transition) was calculated in a way to provide only continuous mass, and thus produced discontinuities in projected area. …"

Line 23 put Re in parentheses

Author response: Changed.

Line 24 what is a free fallspeed

Author response: We changed this sentence and mention that ice crystals were freely falling, compared to other studies in which ice crystals were suspended in the wind tunnel.

"Murakami et al. (1985) studied the $E_c$ between polystyrene latex spheres ($d < 6$ µm) and freely-falling planar ice crystals (1.5 mm $< D <$ 5 mm, and 70 $<$ Re $<$300)."

Line 25 "that were exposed to"

Author response: Corrected.

Line 26 "Although d ranged"

Author response: Corrected.

Page 8  Line 21 "from the Sierra"

Author response: Corrected.

Line 29 and following line both start with Also

Author response: The former changed to "In addition".

Page 9 Line 15 remove hyphens after moderately and heavily

Author response: Changed.

Line 17 add a year to the Magono and Lee citation

Author response: The year (1966) is added,

Page 10 Line 3 the binning intervals description is confusing.

Author response: To avoid the confusion, we added a table (Table 1) to better describe the bin intervals:

"Table 1. Description of the bin intervals of PSD.

| bin number | bin lower point | bin upper point | bin width |
|---|---|---|---|
| 1 | 100 | 200 | 100 |
| 2 | 200 | 300 | 100 |
| 3 | 300 | 400 | 100 |
| 4 | 400 | 500 | 100 |
| 5 | 500 | 600 | 100 |
| 6 | 600 | 700 | 100 |
| 7 | 700 | 800 | 100 |
| 8 | 800 | 900 | 100 |
| 9 | 900 | 1000 | 100 |
| 10 | 1000 | 1200 | 200 |
| 11 | 1200 | 1400 | 200 |
| 12 | 1400 | 1800 | 400 |
| 13 | 1800 | 2400 | 600 |
| 14 | 2400 | 3000 | 600 |
| 15 | 3000 | 4000 | 1000 |

"

Line 7 Do the results change if lightly rimed ice is assumed to be in the rimed ice category?

Author response: Since lightly rimed particles have characteristics more similar to unrimed particles, including them in rimed category would distort the rimed category in a way that the difference between rimed and unrimed mass would be less distinct. Moreover, there are very, very few ice particles in the SCPP dataset (<< 1%) classified as "R1-" type crystals (denoting light riming). Since it was hard to distinguish between R1- and unrimed crystals during SCPP due to the magnification used, R1- was seldom used.

Figure 1 what density is assumed for the solid ice spheres?

Author response: Their density is equal to solid ice density (=0.917 g / m$^3$). We added this to this part of the manuscript:

"… The grey line, corresponding to spherical particles having density equal to 0.917 g m$^{-3}$ (solid ice density), serves as an upper limit to ice particle mass. …"

Lines 14-15 This is shown in Figure 2 and this should be noted

Author response: Although we explained this with more details for dendrites in Fig. 2, we also mentioned this for Fig. 1 to show the similar pattern for all SCPP ice particles. We added a sentence after these lines:

"… We discuss this with more details in Sect. 4.1. …"

Page 11 Line 8 20-30% is less than others get and is less than the value of about a doubling of mass you use later in the paper. How can you explain this?

Author response: This ratio was for a subset of SCPP data that was used to produce Baker and Lawson (2006) m-A power law expression. To avoid confusion, we deleted this sentence. Note that it is consistent with M90 (who estimated 30-40%).

Line 15 perhaps start off by talking about methods that assume constant beta (MG09) and then corroborate their results

Author response: We discuss about this from Page 12 Line 17 to Page 13 Line 4.

Page 12 Eq 3 Can you also plot best fit values assuming D^3? What reduced density values would you get if you assumed D^3 and is this reasonable?

Author response: Assuming that unrimed dendrite mass conforms to the formula $m = \rho_i \pi D^3 / 6$, where $\rho_i$ is ice density, such $\rho_i$ is equal to 0.07 g cm$^{-3}$ for $D = 500$ μm. Note that this assumption would lead to a fit parallel to the ice spheres fit in Fig. 2, with a relatively low coefficient of determination ($R^2 = 0.68$), compared to power law fit with $R^2 = 0.97$.

If we added this fit to Fig. 2, it would overlap with other fits in this figure (we already added unrimed and rimed dendrites and graupel m-D power law fits to this figure). Therefore, we added the above explanations to the manuscript, page 12, starting at line 14:

"… Assuming that unrimed dendrites mass conforms to the formula $m = \rho_i \pi D^3 / 6$, where $\rho_i$ is a reduced density, such $\rho_i$ is equal to 0.07 g cm$^{-3}$ for $D = 500$ μm. Note that this assumption would lead to a fit parallel to the ice spheres fit in Fig. 2, with relatively low coefficient of determination ($R^2 = 0.68$), compared to power law fit with $R^2 = 0.97$. …"

Lines 9-10 Put these statements in better context. Perhaps talk about how bulk models use graupel categories (D^3) and some assume spherical snow, but others assume snow goes as D^2. Then talk about how beta should change in nature and the best way to parameterizing riming snow in models.

Author response: We changed the sentences after Eq. (5) to address this:

"…having an exponent nearly identical to that in Eq. (4) for heavily rimed dendrites. This is contrary to most cloud models that assume different ice categories (snowfall with $\beta \sim 2$ and graupel with $\beta \sim 3$) and an abrupt increase in $\beta$ upon a change in ice category (autoconversion). Based on SCPP observations, it is apparent that the traditional hypothesis that $\beta$ increases with riming is not correct, at least not before the graupel onset. …"

Line 16 Again, if you assumed spherical graupel, what density would it have and could you also plot that m-D curve?

Author response: By assuming that initial graupel mass can be calculated as $m = \rho_i \pi D^3 / 6$ where $\rho_i$ is a reduced density, such $\rho_i$ is equal to 0.18 g cm$^{-3}$ for $D = 500$ μm, which is lower than the $\rho_i$ for heavily rimed graupel in the dry growth regime ($\rho_i = 0.4$ g cm$^{-3}$; Rutledge and Hobbs, 1984; Ferrier, 1994). However, this assumption would produce a fit parallel to the ice spheres fit in Fig. 2, and is poorly fitted to the SCPP R4b and R4c data ($R^2 = 0.67$), compared to power law fit ($R^2 = 0.94$).

If we added this fit to Fig. 2, it would have overlap with other fits in this figure (we already added unrimed and rimed dendrites and graupel m-D power law fits to this figure). Therefore, we added the above explanations to the manuscript, page 12, starting at line 17:

"By assuming that initial graupel mass can be calculated as $m = \rho_i \pi D^3 / 6$ where $\rho_i$ is a reduced density, such $\rho_i$ is equal to 0.18 g cm$^{-3}$ for $D$ = 500 μm, which is lower than the $\rho_i$ for heavily rimed graupel in the dry growth regime ($\rho_i$ = 0.4 g cm$^{-3}$; Rutledge and Hobbs, 1984; Ferrier, 1994). This assumption would produce a fit parallel to the ice spheres fit in Fig. 2, and is poorly fitted to the SCPP R4b and R4c data (R$^2$ = 0.67), compared to the power law fit (R$^2$ = 0.94)."

Page 13 Line 5 "beta as constant during the riming process (until spherical) for both"

Author response: Changed.

Line 8 perhaps use "number distribution" instead of "number density"

Author response: Changed.

Line 9 remove "because it is a function only of D"

Author response: Changed.

Lines 12-18 seem out of place

Author response: We moved this paragraph to Introduction (after the second paragraph in Sect. 1.1).

Line 20 this value is inconsistent with the 20-30% from earlier. Explain.

Author response: As explained previously, the value in this part is correct. The 20-30% value was based on a subset of SCPP that is now removed.

Page 14 Lines 1-3 you can probably just say this is due to errors in classification

Author response: This could be a classification error. However, we often see very large changes in $m_r/m_u$ when only a few ice particles are sampled over a given size-interval. From our experience with this dataset, we feel that large variance in single ice particle mass within a given ice particle type category and size interval is not unusual. Therefore, the sentence was not changed, and it demonstrates the variance that poor statistics can generate.

Page 15 Line 10 Is this also justified because beta is approximately 2?

Author response: We changed the sentence to reflect this point:

"…This assumption can be justified, because $m \propto \dfrac{\alpha}{\gamma} A$ (note that $\beta$, $\delta$ and $D$ does not change during the phase 1 of riming); this can be investigated through future research.."

Line 19 The value of 3.3 will depend on things like rime density. Under what conditions does using this value work?

Author response: This should be generally valid for planar ice crystals. While "rime density" may vary considerably on a single particle basis, these results are for a population of graupel particles where these single-particle differences are averaged to provide a bulk value for the population.

Page 17 Line 5 again the value of 2.4 will depend on a lot of variable. When do you expect the model to break down?

This should be valid for the SCPP dataset. It is possible that graupel formed in Sierra Nevada snowstorms experiences more "wet growth" (i.e. cloud droplets do not immediately freeze upon impacting the ice crystal) whereas in Rocky Mountain snowstorms, graupel experiences primarily dry growth (droplets freeze immediately upon impact, resulting in lower "densities"), but this is speculation that cannot be tested within this SCPP dataset. Moreover, we do not know how much this might affect our results; we feel it is best not to speculate on this.

Page 22 Line 5 doesn't increasing size also lead to increasing Re?

Author response: In theory, Re also depends on D. However, when an ice particle falls in a mixed-phase cloud, riming causes an increase in its fallspeed, but not its maximum dimension.

Line 11 "as a conservative underestimate"

Author response: Changed.

Line 19 perhaps "number distribution"

Author response: Changed.

Line 21 it is assumed to be negligible, not zero

Author response: We changed this sentence:

"… Note that the cloud droplet sedimentation velocity $v(d)$ is negligible compared to the ice particle fallspeed $V(D)$ and was neglected in the similar equation by Heymsfield (1982), M95, and Zhang et al. (2014). …"

Page 23 Line 7 "microphysical and therefore optical"

Author response: Many models use separate microphysical and optical parameterizations, and the variable in one parameterization might not depend on the other parameterization.

Page 24 Line 11 is it really d^13?

Author response: Yes, because $v = 9$, and from Eq. (25) the dimension of n(D) is $d^9 / d^{13} = d^{-4}$.

Page 25 Line 11 should this be "representing ice particle m and D"?

Author response: Our methods calculated both projected area and mass. We used "mass" and "projected area" instead of "m" and "A" to avoid confusion.

Page 26 Line 1 change SGM to simple growth model

Author response: SGM stands for snow growth model.  SGM has now been replaced by "snow growth model".

[Figure]

Fig. 1.

[revised manuscript text omitted]

---

## Author Response (AR2)

**Response to the referee comments on the manuscript:**

Title: Growth of ice particle mass and projected area during riming
By: Erfani, Ehsan; Mitchell, David
Article reference: acp-2016-455, revised submission

We wish to thank the referees for their detailed and helpful comments on our paper. As you will see below we have responded to all of the comments with revisions designed to address the concerns of the referees. In the following response, the original referee comments appear in black and our responses appear in blue and are labeled "Author response:"

**Referee comments:**

**Anonymous Referee #1:**

There is no comment from referee #1

**Anonymous Referee #2:**

1. Equations 6-8 are similar to the method of Morrison and Milbrandt (2015) and this needs to be pointed out.

Author response: The reason for such similarity between our work and Morrison and Milbrandt (2015) is that they used our preliminary results of SCPP data analysis (Mitchell and Erfani, 2014) to develop their model, in addition to observations of Rogers (1974). This is explained in page 14, starting at line 21:

"All these observations are in agreement with the experiment of Rogers (1974) in which $\beta$ was similar for unrimed and rimed snowflakes. The results of Rogers (1974) were used in the modeling work of MG08 and Morrison and Grabowski (2010) to support the assumption that riming does not change $\beta$ for planar ice crystals. Morrison and Milbrandt (2015) used a similar assumption based on the observations of Rogers (1974) and Mitchell and Erfani (2014), and they explained that the reason for the conservation of $\beta$ during riming is the fact that D does not significantly change by riming while m does increase significantly."

2. In sections 5.2 (Hexagonal columns) and section 7 (conclusions) it should be pointed out that Jensen and Harrington (2015) calculate collision efficiencies for prolate ice based on aspect ratio, which could be used in model.

Author response:

We added explanations to clarify this.

Sect. 5.2, page 23, starting at line 9:

"To the best of our knowledge, there is not any practical $E_c$ equation for such crystals in the literature, suitable for use in cloud resolving models, except for the study of JH15 that calculated $E_c$ for prolate spheroids based on their aspect ratios."

Sect. 7, page 28, starting at line 15:

"Prior to this work, there was only one rigorous practical method for calculating the droplet size-dependence of $E_c$ for use in models. As described in JH15, this method when applied to prolate spheroids modified the equation from Beard and Grover (1974) for spherical raindrops in steady flow, and calculated $E_c$ between a cloud droplet and a prolate spheroid based on the spheroid aspect ratio."

3. In section 7 (conclusions) line 10, would models that use multiple categories still use multiple categories if they implemented your method? This is not clear.

Author response:

This is now addressed under conclusions, page 28, starting at line 12:

"By using the method introduced in this study, models may still use multiple categories (e.g. ice crystal, rimed particle, graupel), but within each category the rimed mass fraction can gradually increase, thus preventing an abrupt change in ice particle attributes between categories."

The table needs units.

Author response: Units are added to the table.

[revised manuscript text omitted]